# Systematic data quality assessment of electronic health record data to evaluate study-specific fitness: Report from the PRESERVE research study

Hanieh Razzaghi[1]*, Amy Goodwin Davies[1], Samuel Boss[1], H. Timothy Bunnell[2], Yong Chen[3], Elizabeth A. Chrischilles[4], Kimberley Dickinson[1], David Hanauer[5], Yungui Huang[6], K. T. Sandra Ilunga[1], Chryso Katsoufis[7], Harold Lehmann[8], Dominick J. Lemas[9], Kevin Matthews[10], Eneida A. Mendonca[11], Keith Morse[12], Daksha Ranade[13], Marc Rosenman[14], Bradley Taylor[15], Kellie Walters[16], Michelle R. Denburg[3,17,18], Christopher B. Forrest[1,17], L. Charles Bailey[1,17]

1 Applied Clinical Research Center, Departments of Pediatrics and Biomedical and Health Informatics, Children's Hospital of Philadelphia, Philadelphia, Pennsylvania, United States of America, 2 Biomedical Research Informatics Center, Nemours Children's Hospital, Wilmington, Delaware, United States of America, 3 Department of Biostatistics, Epidemiology, and Informatics, Perelman School of Medicine at the University of Pennsylvania, Philadelphia, Pennsylvania, United States of America, 4 Department of Epidemiology, College of Public Health, University of Iowa, Iowa City, Iowa, United States of America, 5 Department of Learning Health Sciences, University of Michigan Medical School, Ann Arbor, Michigan, United States of America, 6 IT Research and Innovation, Nationwide Children's Hospital, Columbus, Ohio, United States of America, 7 Division of Pediatric Nephrology, University of Miami Miller School of Medicine, Miami, Florida United States of America, 8 Biomedical Informatics & Data Science Section, Johns Hopkins School of Medicine, Baltimore, Maryland, United States of America, 9 Department of Health Outcomes & Biomedical Informatics, University of Florida, Gainesville, FLorida, United States of America, 10 Analytics Research Center, Children's Hospital of Colorado, Aurora, Colorado, United States of America, 11 Division of Biomedical Informatics, Cincinnati Children's Hospital Medical Center, Cincinnati, Ohio, United States of America, 12 Division of Pediatric Hospital Medicine, Stanford University School of Medicine, Stanford, California, United States of America, 13 Biostatistics, Epidemiology, and Analytics in Research (BEAR), Seattle Children's Hospital, Seattle, Washington, United States of America, 14 Department of Pediatrics, Ann & Robert H. Lurie Children's Hospital, Chicago, Illinois, United States of America, 15 Clinical and Translational Science Institute, Medical College of Wisconsin, Milwaukee, Wisconsin, United States of America, 16 Translational and Clinical Sciences Institute, University of North Carolina at Chapel Hill, Chapel Hill, North Carolina, United States of America, 17 Department of Pediatrics, Perelman School of Medicine at the University of Pennsylvania, Philadelphia, Pennsylvania, United States of America, 18 Division of Nephrology, Children's Hospital of Philadelphia, Philadelphia, Pennsylvania, United States of America

* razzaghih@chop.edu

## Abstract

Study-specific data quality testing is an essential part of minimizing analytic errors, particularly for studies making secondary use of clinical data. We applied a systematic and reproducible approach for study-specific data quality testing to the analysis plan for PRESERVE, a 15-site, EHR-based observational study of chronic kidney disease in children. This approach integrated widely adopted data quality concepts with healthcare-specific evaluation methods. We implemented two rounds of data quality assessment. The first produced high-level evaluation using aggregate results from a distributed query, focused on cohort identification and main analytic requirements. The second focused on extended testing of row-level data centralized for analysis. We systematized reporting and cataloguing of data

**Data Availability Statement:** The results reported in this manuscript are based on detailed individual-level patient data compiled by members of PCORnet. Due to the high risk of reidentification based on unique patterns in the clinical data, even when demographic identifiers have been removed, patient privacy regulations prohibit us from releasing the individual-level data publicly. However, we have made our materials available as well as the code underlying the analysis, which are publicly available and can be accessed here: https://github.com/PRESERVE-Coordinating-Center/preserve_ssdqa Please direct requests to access the data, either for reproduction of the work reported here or for other purposes, to preserve@chop.edu.

**Funding:** This work was funded by an award from the Patient-Centered Outcomes Research Institute (https://www.pcori.org) number RD-2020C2-20338, with MD and CBF as principal investigators. The funder had no role in study design, data collection and analysis, decision to publish, or preparation of the manuscript.

**Competing interests:** One author (MD) reports funding from Mallinckrodt, Inc. for development of the Glomerular Learning Network (GLEAN) for the study of kidney disease in children.

quality issues, providing institutional teams with prioritized issues for resolution. We tracked improvements and documented anomalous data for consideration during analyses. The checks we developed identified 115 and 157 data quality issues in the two rounds, involving completeness, data model conformance, cross-variable concordance, consistency, and plausibility, extending traditional data quality approaches to address more complex stratification and temporal patterns. Resolution efforts focused on higher priority issues, given finite study resources. In many cases, institutional teams were able to correct data extraction errors or obtain additional data, avoiding exclusion of 2 institutions entirely and resolving 123 other gaps. Other results identified complexities in measures of kidney function, bearing on the study's outcome definition. Where limitations such as these are intrinsic to clinical data, the study team must account for them in conducting analyses. This study rigorously evaluated fitness of data for intended use. The framework is reusable and built on a strong theoretical underpinning. Significant data quality issues that would have otherwise delayed analyses or made data unusable were addressed. This study highlights the need for teams combining subject-matter and informatics expertise to address data quality when working with real world data.

## Author summary

It is important to understand how the data used in a study meet the needs of analyses, especially when the study is reusing data collected for clinical care, such as electronic health records, and repurposed for analyses, rather than collecting new data. This effort, which we term data quality assessment (DQA), currently varies widely across studies. We describe how a process for systematic data quality assessment that is based on principles from informatics theory and practices from healthcare research can be applied to large, multicenter studies, using a study of chronic kidney disease as an example. We performed DQA in two rounds, first remote and then on the analytic dataset, showing that each approach identifies many data quality issues that can affect study results. Using our results, institutional data teams were able to fix over 120 issues, including 2 sites that would otherwise have needed to drop out of the study. The study team used the results to make final definitions of analytic variables, and can account for other intrinsic data problems through statistical methods. This standardized approach, including design, organized results visualization, and interaction with data providers and statisticians, can be replicated for other clinical studies.

## Introduction

Secondary use of electronic health records (EHRs) to conduct clinical research has gained momentum in recent years [1–4]. Multi-institution research networks using EHRs as a data resource vary in infrastructure and governance but share the goals of using clinical data for knowledge discovery and improving health outcomes across the lifespan. Because researchers frequently rely on data collected for health care delivery, a common challenge is ensuring data are of sufficient fitness to minimize analytic risks such as misclassification, bias, undetected stratification, or other confounding, and to maximize the generality of results [5–10]. This effort is variously described as data quality assessment, data cleaning, exploratory data analysis,

preparatory investigation, or data curation. Practices differ widely depending on the background and experience of investigators and on the structure of the study. This extends to dissemination of results as well, with some studies reporting little more than a descriptive profile of the final dataset while others include more detailed evaluation of how data characteristics may affect study results, typically in supplemental materials.

Informaticians involved in the infrastructure of networks have developed standing data quality programs that detect basic anomalies but are typically agnostic of clinical context or specific research questions [11–13]. These programs usually conform to the harmonization framework established by Kahn *et al.* [14] in constructing data quality checks for *completeness*, *plausibility*, or *conformance*. These terms serve as a common reference for network informaticians and researchers to broadly describe data quality. Most multi-site networks such as PCORnet [9], AllofUs [5], N3C [15], and the OHDSI community [16] have adopted these terms to categorize and describe their data quality checks, and as of this writing, the paper has been cited in 181 times. However, while these terms are useful to frame data quality dimensions for cataloging, they do not provide the tools for formulating checks and lack context-independent specifications required for true interoperability between networks. Further, network checks rarely evaluate whether data are *fit-for-use* in study-specific contexts, that is, can provide accurate and representative values for analytic variables. A small number of study-specific data quality frameworks have been proposed [17,18], but they have not been rigorously evaluated in multi-institutional data and do not provide specific tools for reuse or more granular terminologies to catalog checks, instead relying on multiple iterations of domain expert input and descriptions of data quality checks.

Similarly, standard practice during study execution includes some interrogation of data quality, typically for missingness and facial implausibility of key variables. Significantly, much of the statistical literature focuses on addressing specific problems related to confounding or misclassification [19], or focus on solving a specific data quality problem, such as detecting outliers in blood pressure trajectories [20]. While valuable, these tests do not detect higher-order problems, such as shifts over time, differences in clinical or data capture practice across study institutions, non-demographic internal stratifications (*e.g.*, by clinician specialty), or subgroup-specific anomalies. They are also intended to resolve a well-characterized anomaly in the data, and do not necessarily inform decisions about study design, variable definitions, or analytic priorities, in the same way that study-specific frameworks can address. The lack of such systematic checks assessing fitness for analysis often leads to delays when conducting studies, as significant data quality issues are discovered late in analyses. Of greater consequence, unanticipated issues can cast doubt on the integrity of the data overall and lead investigators to discard useful information or even abandon their study question. Several EHR-oriented data quality frameworks have been proposed [18,21–23],with varying degrees of complexity and specificity. However, most clinical research studies lack resources for complex development or implementation needed for comprehensive data quality programs. Moreover, study-specific data quality assessment (SSDQA), that is, testing focused on the particular analytic needs of that study, cannot by its nature simply apply checks developed elsewhere in a straightforward way. Finally, evaluation of SSDQA for research based on standardized data in large networks has been limited.

We address here this need for investigating data quality integrated into research studies by systematically constructing and demonstrating the effectiveness of *semantic* or *study-specific* data quality checks in the Preserving Kidney Function in Children With Chronic Kidney Disease (PRESERVE) study of chronic kidney disease in children. This study makes use of data from PCORnet [24], the national patient-centered research network, in which member institutions standardize data to a common data model [25] to support a range of research studies. In

addition to this network context, the large scale, diversity of institutions and data platforms, and range of study variables provide an opportunity for testing several dimensions of SSDQA. Further, all institutional data sources had successfully completed standard PCORnet data curation [9], which addresses general structural aspects of data quality, providing us with the opportunity to evaluate the incremental utility of SSDQA.

We build on the SSDQA framework described by Razzaghi *et al.*[26], which focuses on aligning informatics data quality principles (e.g., *completeness*, *conformance*) with study-specific constructs (e.g., *cohort definitions*, *prioritized variables*), methods of evaluation (e.g., *frequency distributions*, *visualizations*), and clinical or epidemiological concepts (e.g., *utilization*, *clinical care patterns*). We demonstrate application to PRESERVE's EHR-based data to identify key data quality issues early in study execution. Our goals are twofold. First, we provide evidence for the value of SSDQA by showing insights into the data that can be used to guide study design and drive measurable improvements in study execution. Second, we provide a detailed record of the PRESERVE SSDQA process to facilitate reuse in other studies.

## Materials and methods

### Study setting

PRESERVE is a large-scale observational study that integrates data from 15 health system participants in the large, multi-institutional EHR research network PCORnet [24], to evaluate whether different approaches to controlling high blood pressure in children with chronic kidney disease (CKD) may slow the loss of kidney function. The study uses EHR data, staged at institutions as part of the PCORnet infrastructure, augmented with area-level measures and kidney disease registry data. Participating institutions include children's hospitals as well as pediatric services in all-age hospitals, with a range of population sizes, specialty focus, and health information technology infrastructure represented. It incorporates an aim to enhance the PCORnet Common Data Model (CDM) for pediatric and rare kidney disease, which included the objectives of performing SSDQA and addressing issues for data quality optimization. The PEDSnet Data Coordinating Center (DCC) developed, coordinated, and executed the SSDQA as part of their role as the Coordinating Center for PRESERVE. The work presented here was approved as part of the PRESERVE study protocol by the Children's Hospital of Philadelphia Institutional Review board (Protocol 21–018814), which serves as a central IRB for PRESERVE.

EHR-derived data for this study were obtained from PCORnet [9,24] core data at each institution. These data are extracted quarterly from clinical source systems and transformed to PCORnet Common Data Model (CDM) specifications. Before the PRESERVE queries, each institution's data successfully completed standard PCORnet Data Curation, required for network participation [9]. Additional PRESERVE-specific data elements were extracted at institutions for the study cohort outside the quarterly data extraction, transformation, and loading (ETL) process, and integrated into the structure of the PCORnet CDM for further analysis. All data used as input to SSDQA conformed to the definition of a Limited Data Set in the Health Insurance Portability and Accountability Act's Privacy Rule [27], incorporating complete dates and neighborhood-level geocodes. All other direct identifiers were either omitted or replaced with study codes. All data were maintained in secure databases with access limited to study personnel with a need to analyze the data.

### SSDQA Check Construction

Following the framework described by Razzaghi *et al.* [26], we undertook four steps (Fig 1) to develop and annotate SSDQA checks, described below. Several key principles guide SSDQA

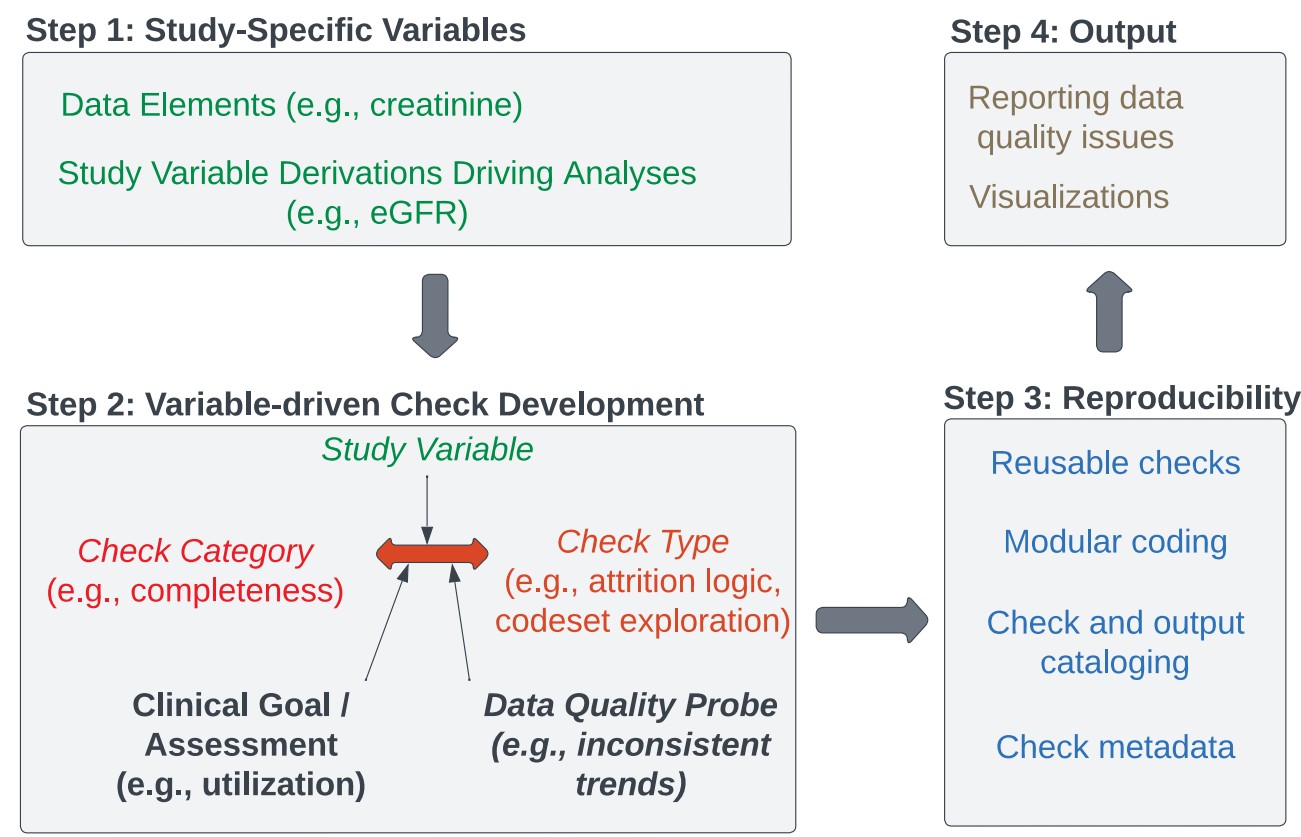

**Fig 1. Development of PRESERVE SSDQA Checks.** The framework for SSDQA design was applied to the PRESERVE study, pairing variables (Step 1) with well-established data quality terms and check types and additional descriptive metadata (Step 2), focusing implementation on technical and methodological reproducibility (Step 3), to produce interpretable output (Step 4).

development. First, mapping study variables to specific data elements is necessary to determine the study's data quality requirements. These mappings may be one to many for variables derived from multiple types of data. Second, check development benefits from combining consideration of *check category*, a goal-directed attribute such as completeness or plausibility, and *check type*, a method-directed attribute such as measuring data density or matching expected laboratory testing to diagnosis. Third, SSDQA should be both technically reproducible, with an emphasis on modular coding to allow reuse of checks across variables, and methodologically reproducible, with documentation for thorough data quality cataloging. Finally, results should be actionable, either highlighting an ETL error or informing analysis planning.

*Step 1*: *Study-Specific Variables*. We began by identifying variables associated with the study's scientific aims and analysis plan, including eligibility criteria, and mapped them to the semantics of the PCORnet data model. Importantly, we distinguished between *data elements*, which are single values represented in the PCORnet CDM, and *study variables*, which are the combination of data elements with logic for their application. For example, estimated glomerular filtration rate (eGFR) is a study variable, and can be derived from age, height, and either serum cystatin C or serum creatinine. We suspected that the former laboratory test would be associated with greater missingness given that it has been widely implemented only in recent years, and focused DQA testing accordingly. We evaluated the importance of data elements

**Table 1. Check and issue characteristic metadata for PRESERVE SSDQA.**

| Check Category | | | | |
|---|---|---|---|---|
| completeness | conformance | plausibility | concordance | consistency |
| **Check Type** | | | | |
| attribution | attrition step check | clinical correlation | clinical fact relationships | clinical fact density |
| clinical thresholds | code utilization | frequency distributions | specialty correlation | summary statistics |
| temporal relationships | trends over time | utilization thresholds | variable stratifications | |
| **Clinical Goal** | | | | |
| clinical care | diagnostic evaluation | epidemiologic distribution | utilization | |
| **Data Quality Probe** | | | | |
| anomalous event sequencing | anomalous values | data represent-ation error | Inconsistent trends | |
| **Data Quality Issue** | | | | |
| atypical code distribution | atypical numeric distribution | discordant values | high counts | high spike |
| low counts | low spike | mapping error | missingness | outlier values |

and study variables to implement the following prioritization: eligibility criteria > main exposures and outcomes > covariate; multiple analyses > single.

*Step 2*: *Check Development*. Next, we used five data quality categories to guide check development and application: concordance (agreement of data elements across multiple tests, such as diagnoses and medications), completeness (presence of sufficient clinical information, or facts, to support accurate analyses), plausibility (whether clinical information follows expected patterns based on prior data or subject matter expertise), consistency (reliability of values across repeated measurements), and conformance (alignment with data model). Because all data contributors had completed regular PCORnet data curation [9], we expected that basic completeness and conformance had been addressed, and focused on study-specific applications. We connected a check category (*e.g.*, completeness) with a methodologic *check type*, (*e.g.*, attrition logic, codeset exploration, variable stratifications, *etc.*) for different sets of variables (*e.g.*, blood pressure, urine protein, urine creatinine, *etc.*)[26]. While the check category reflects broad data quality principles, and has been the primary focus in data quality research in informatics, the check type specifies concrete analytic output. We specified a *clinical goal* (*e.g.*, utilization) and *data quality probe* to ensure sufficient metadata and dimensions for check specification. The output of this step provided adequate information to formulate and document checks. Table 1 shows the complete lists of check category, check type, clinical goal/assessment, and data quality probe values applied for this study.

The framework we describe here comprises reusable tools for all studies, and the specified checks as an example of their application. Table 2 shows examples of five fully specified checks, with each serving as an example of how the four dimensions can be applied to a variable in the study to further specify a check. While we defined these checks for this specific application, we anticipate that their reuse across different contexts can be intuitive. For example, in the first row of Table 2, we derived a check by combining the dimensions: completeness, trends over time, utilization, and missingness. In this study, we specified the check application to measure the proportion of patients with at least one serum creatinine measurement per year. However, in a cohort of diabetes patients, the study team may wish to apply the check to patients with at least one HbA1c measurement or alter the check slightly to focus on patients with an A1c value > = 6.5. Alternatively, they may wish to instead change to a focus of examining [data quality probe = inconsistent trends] and design the check to identify time points that are +/- 2 SD from the mean.

*Step 3*: *Reproducibility*. Our intent in DQA construction was both systematization as well as reproducibility. We addressed the former through structured and methodical process of check

**Table 2. Examples of Check Specifications.**

| Check Category | Check Type | Clinical Goal | Data Quality Probe | Check |
|---|---|---|---|---|
| completeness | trends over time | utilization | missingness /data representation error | proportion of patients with at least one sCr measurement per year; standard of care for CKD |
| concordance | clinical fact relationships | clinical care | missingness | systolic and diastolic BP measurements present on the same day (more granular resolution not supported) |
| consistency | clinical correlation | clinical care | Inconsistent trends | correlation of systolic and diastolic blood pressures |
| conformance | frequency or density distributions | clinical care | missingness | proportion of urine protein values with unit type mapped |
| consistency | trends over time | clinical care | anomalous event sequencing | eGFR median before and after CED, years from cohort entry; assess whether trend is pathologically consistent with CKD |
| plausibility | frequency or density distributions | clinical care | anomalous values | urine protein: quantitative distribution of results; assess whether results reflect expected values for CKD patients |

development, application, and output, as well as site communication and reporting. To improve reusability and hence reproducibility across studies, we adopted software development best practices in the writing of our checks, such as emphasis on documentation, structured source code, and segmentation of code [28–30].

*Step 4*: *Output*. For each check, visualizations were constructed for dual use by informatics teams managing data and clinical research teams conducting analyses. We reviewed the results at the coordinating center with a team of data scientists and clinical informaticians for further refinement of the output. Annotated results were collected into reports distributed to both data teams and clinical investigators.

## SSDQA Check Sequencing

Because PCORnet operates as a distributed network, and sharing of detailed data is limited to analytic datasets, we divided the PRESERVE SSDQA effort into two distinct rounds of data quality assessment (DQ1 and DQ2) and efforts at resolution before the start of research analyses. DQ1 used a fully distributed query returning aggregate data only. The output was intended to guide the development of the analytic dataset and focused on eligibility criteria and main analytic variables, and corresponding data elements, with a focus on value distributions. After feedback to institutional informatics teams and ETL revision, DQ2 used a centralizing query returning row-level data to the study coordinating center, where SSDQA took place. This round applied data quality checks across multiple clinical dimensions, providing the ability to more robustly assess fitness for intended analyses. DQ2 also re-executed the DQ1 query to track any improvements in output as a result of detailed feedback to sites. To incorporate ETL improvements from the DQ2 feedback and revision, we performed a final row-level query, with SSDQA again performed at the coordinating center on the returned results (DQ2R). This returned dataset formed the final analytic data for the study.

For both the re-execution of DQ1 and DQ2, the PRESERVE Coordinating Center (CC) reviewed all results and determined whether there were measurable improvements for each issue and annotated reasons based on feedback from sites if no improvement was made. These annotations were grouped into broader categories following review by the PRESERVE CC team.

## SSDQA Check Execution

In each round, code for execution against the PCORnet CDM, termed a query, was developed in the SAS programming language by the PRESERVE CCC to run on the and was distributed

to all participating institutions. This was done to follow standard PCORnet practice for interacting with data in the network. For DQ1, this query executed the data quality (DQ) checks, and returned resulting count data to the Coordinating Center. Subsequent queries identified patients who met cohort eligibility criteria and retrieved row-level data that would be used to construct study variables. The resulting datasets were returned to the Coordinating Center, and DQ checks were executed on these data.

We developed a *check catalog* to document the process of check development and annotated checks with check_num (unique check ID), summary, check type, domain, variable, check category, data quality probe, and clinical goal. The summary was derived from combining the check category and check type with the descriptive metadata from the data quality probe and clinical goal, as applied to specific variable names. Complete catalogs of checks and accompanying annotation are provided in S1 and S2 Tables.

Results were reviewed in both group virtual meetings and in communication with institutional teams. Interpretation of results and investigation of potential causes included input from clinical informaticians, clinical subject matter experts, and data analysts. Results could be classified as a data quality issue if they differed significantly from that of most institutions in the study, if they were assessed as substantially different from practice standards by clinical SMEs, or if they appeared to reflect a systematic shift characteristic of an error in ETL code (*e.g.* miscoding of units). While each issue reflects a unique set of causes, we have included in S3 Table a set of common considerations in addressing different DQ issue types. Data quality problems determined to be ETL limitations were assigned to be addressed at the next data refresh. Source data limitations were assessed to determine what analytic adjustments were needed.

## SSDQA Evaluation

We constructed an *issue catalog* to annotate issues detected by the checks and support systematic evaluation of data quality issues. The catalog comprised issue id (unique issue ID), priority, clinical or data domain, element (specific study variable or data element), dq issue (value set listed in Table 1), info (additional information for reviewer), and report reference (fig number in report). Clinical domain was assigned based on the type of clinical entity the data quality check intended to measure, while element focused on specific variables. In order to facilitate dialogue and document improvements, we set up a GitHub organization for this study (https://github.com/PRESERVE-Coordinating-Center) where we shared codesets and common definitions used in the DQ reports. In addition, we created a separate private repository for each participating institution where we shared issues from the issue catalog for that institution.

For each round of DQ, we created a report that collected results into visualizations including tables and figures that incorporated all contributors, anonymizing names of institutions. This allowed the user to more easily interpret data quality issues overall and their potential impact on the study, and also facilitated benchmarking across institutions. Further, two members of the PRESERVE Coordinating Center (authors HR and CB) reviewed and collaboratively assigned every issue to one of four priority levels using the following criteria: (1) urgent, which was assigned to issues where analyses would have not been possible using the affected data, and were related to cohort attrition or missingness or highly atypical distributions of major variables; (2) high, which identified missingness of variables not foundational to the study but nevertheless potentially disruptive to specific analyses; (3) medium, which were related to abnormalities that were more likely to be characteristics of source data or variable metadata; (4) low, which were also related to abnormalities or characteristics of the source data but were likely to have minimum impact on study analyses.

To meaningfully summarize our data quality results, we categorized issues into broader themes. Check results were evaluated to classify the specific data quality finding, considered in the context of data quality issue value set (Table 1), domain, variable importance, and potential cause. Categories were proposed by author HR and assigned to integrate study analytic requirements, and reviewed with the analysis team. The issue catalog was reviewed in meetings to ensure that individual issues could be assigned to one or more categories. Final assignments were reviewed by authors HR and CB, who concurred in each.

## Results

### Study setting

Fifteen health systems contributed data to the PRESERVE cohort, each completing the data quality assessment process. In the final cohort of 178,928 patients, the number from a single health system ranged from 2,657 (1.5%) to 24,863 (13.9%) with a median of, 9,074 (5.1%); IQR 6,924.5 (3.9%)– 16104 (9.0%). All were affiliated with academic institutions and served as teaching hospitals. Eight were freestanding children's hospitals, while the remainder were all-age health systems. All were members of PCORnet, and informatics staff participating in the study also supported overall PCORnet data management.

### Data quality findings

The DQ1 analysis, using distributed queries returning aggregate results from each institution, included 79 data quality checks (S1 Table). These were distributed across data domains, with 8 checks focused on anthropometrics, 8 on vital signs, 8 on diagnoses/conditions, 16 on laboratory tests, 24 on medications, 8 on procedures, 4 on other eligibility criteria, 1 on secular trends in cohort entry, 1 on specialty, and 1 on follow-up duration. Checks fell into four categories: 29 focused on evaluation of completeness, 14 of concordance, 17 of conformance, and 34 of plausibility. As the counts indicate, the results of a single check can be used to assess different aspects of data quality through use of distinct postprocessing. The report comprised 69 figures and 33 tables and identified 115 issues. After review, 10 (9%) issues were designated as urgent, 26 (23%) as high priority, 26 (23%) as medium, and 53 (45%) as low.

The DQ2 analysis, executed on row-level data for potential cohort members centralized at the Coordinating Center, contained 65 unique checks (S2 Table), extending the evaluations from the prior round. These interrogated thirteen domains, with three checks covering two domains each: 1 focused on anthropometrics, 13 on vital signs, 9 on diagnoses/conditions, 8 on geographic variables, 13 on laboratory testing, 4 on derivations for study analyses,4 on medications, 5 on procedures, 2 on eligibility criteria, 2 on secular trends in cohort entry date, 2 on specialty, 1 on cross-domain testing of clinical facts, and 4 on visit characteristics. These spanned five check categories: 26 checks addressed completeness, 5 concordance, 8 conformance, 9 consistency, and 27 plausibility. The DQ2 report comprised 55 figures and 8 tables and identified 157 issues. After review, 8 (5%) issues were designated as urgent, 72 (46%) as high priority, 74 (47%) as medium, and 3 (2%) as low.

Table 3 shows the themes assigned to issues, reflecting their potential impact on study analyses. This ranged from immediate threats to analyses or participation, to the potential for future covariate adjustment. As an example of the former, several institutions did not extract all eligible nephrology specialty visits during their initial ETL process from clinical data systems to research data, leading to cohort sizes that were too small and potentially misrepresentative of local clinical practice (Theme: Anomalies affecting eligibility). Other major effects included the data-quality-driven decision to use serum creatinine in preference to serum cystatin C as the primary test of kidney function, despite the greater clinical accuracy of the latter,

**Table 3. Description of Themes Identified from Data Quality Issues.**

| Theme | Description | Examples of Data Quality Issues | Study Impact |
|---|---|---|---|
| **Anomalies affecting eligibility** | Identifies DQ issues affecting cohort eligibility criteria or pool of eligible patients | Nephrology specialty not represented in data | • Threat to study validity if cohort cannot be finalized or if cohort is biased |
| **Variation in time of cohort entry** | Measures availability of data as well as clinical content of data for the duration of the study | Several institutions had shorter duration of follow up and sparse data prior to 2015 | • Analyses must account for different available lengths of follow up time<br>• Potential for secular shifts in clinical practice or coding affecting analyses |
| **Variation in clinical utilization** | Detects whether utilization patterns align with expectations of cohort | *Other Ambulatory* visits as well as specialty utilization, differed between institutions, which reflects differences in how visits are classified | • Filtering events based on visit type may produce biased results and miss clinical events because of a misclassified visit type<br>• Adjusting for comorbidities will be important to account for patient-case mix |
| **Missingness of major variables** | Measures degree of missingness across critical variables | Several sites had missing serum cystatin C testing | • May require additional ETL work or adjustment of study variable definitions (e.g., using serum creatinine in place of serum cystatin C) |
| **Duplication of values** | Identifies variables that may be anomalous because of duplicate values | One site had anomalously high number of height measurements per patient due to duplicate measurements extracted during ETL process | • Utilization measures will be biased in patients with duplicate data<br>• Duplicated values may be associated with different dates or clinics |
| **Variation in code utilization** | Measures whether the concepts used for a particular variable align with expectations | Procedure codes for dialysis used across sites do not align with specificity of nephrologist clinical practice | • Analyses must account for site heterogeneity in expressing similar concepts<br>• Departure from expected values may require additional data source (e.g., USRDS data for dialysis codes) |
| **Implausible value detection** | Detects quantitative values that are implausible | One site's defaulted value of 9999 for missing laboratory results created a large gap between mean and median lab values | • Artifactual values must be recognized and removed from study analyses |
| **Implausible temporal trends (DQ2 Only)** | Detects anomalous time trends | Several sites had inconsistent utilization spikes in their data | • Analyses must account for potentially incomplete or unrepresentative data for a particular time point |
| **Anomalous distribution of results (DQ2 Only)** | Results from laboratory tests produce anomalous distribution or output | Several sites had urine protein values that were implausible or likely not representative of their source system | • Lack of plausible lab results will require either advanced imputation methods or removal from the study analysis |
| **Event sequencing anomalies (DQ2 Only)** | Detects implausible or anomalous sequencing of clinical events | Patients across several sites have a lower eGFR prior to cohort entry in earlier years of the observation period than later years | • The eGFR difference between prior to and post- cohort entry is smaller in more recent years, which may reflect changes in clinical practice |
| **Comparatively anomalous institutional clinical values (DQ2 Only)** | Identifies sites with data that deviate from others, though the underlying cause may reflect clinical practice | Two sites with markedly more eGFR values per patient than others, though not necessarily anomalous at face value | • Site heterogeneity in case mix or clinical practice will yield different results<br>• Study investigators must ensure that analyses apply appropriate selection criteria and variable adjustment |

due to its poor data capture (Theme: Missingness of major variables), as shown in Fig 2. Similarly, due to complex segmentation of care that made it challenging to identify long-term dialysis in available EHR systems, the study elected to rely instead on the United Stated Renal Data System (USRDS), which maintains a Medicare claims dataset that reflects the program's near-complete coverage of chronic dialysis services in the US. This is reflected in Fig 3, which highlights the heterogeneity of chronic dialysis code selection and frequency in EHR systems (Theme: Variation in code utilization). Face validity was low as well, with overall frequency below the rates of chronic dialysis expected by nephrology subject matter experts. Overall, the DQ1 themes that emerged were often addressable, if not fully resolvable, and related to ETL issues or unresolvable problems with the underlying data.

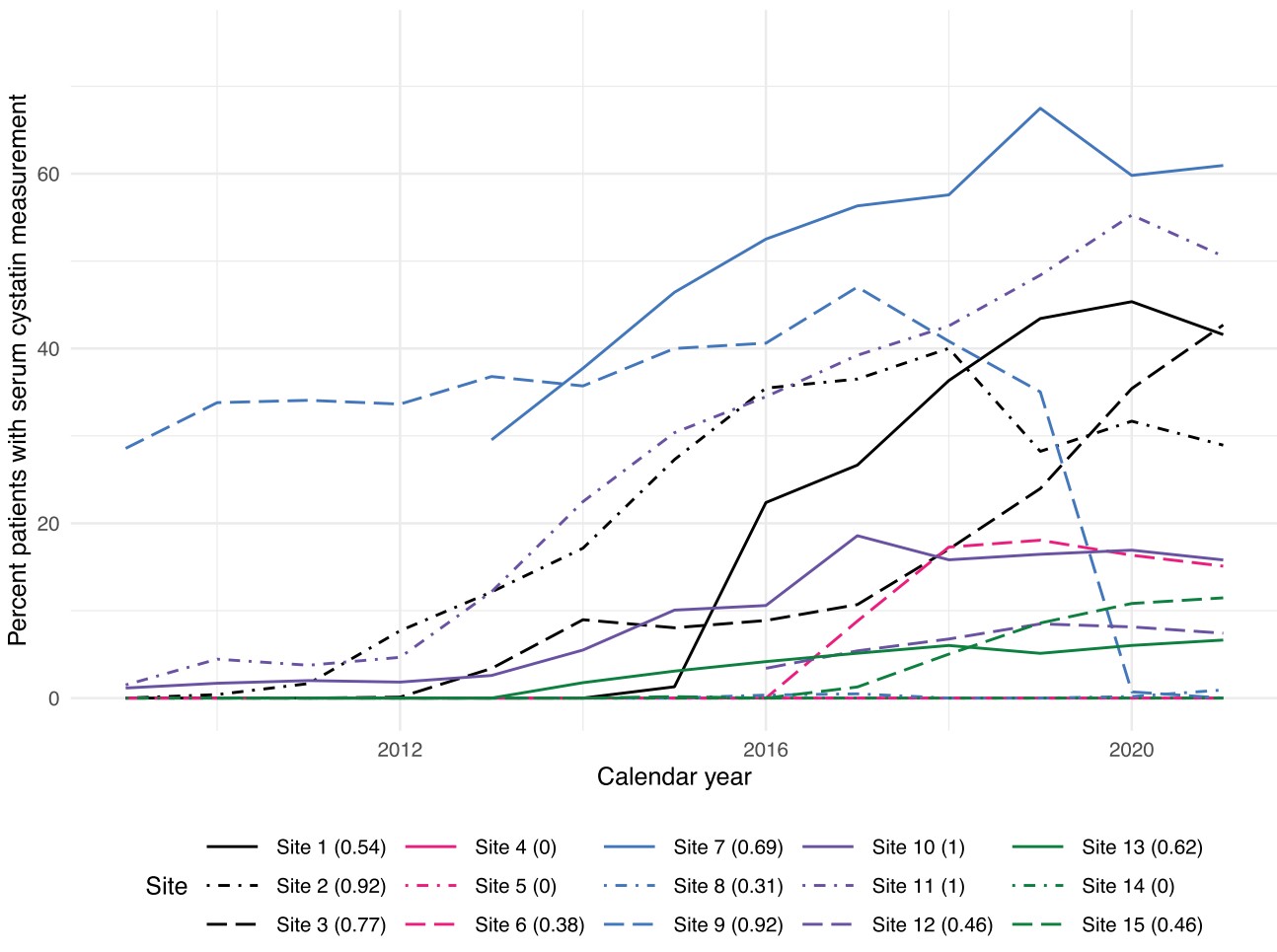

**Fig 2. Data Quality Theme: Missingness of Major Variables.** *Percentage of patients with serum cystatin C value by calendar year.* For each year in which an institution had data for eligible patients, the proportion of patients from that year who had at least one result for a serum cystatin C test, which measures GFR more accurately than creatinine, is shown.

DQ2 included evaluation of time-dependent treatment patterns, comparison of disease trajectories over time, and use of unstructured source values to compensate for missingness or implausible mappings in study variables. As a result, the checks were more involved and included more complex output and data quality themes. For example, estimated glomerular filtration rate (eGFR) is the fundamental measure of kidney function for the study. DQ1 assessed missingness in the underlying data elements to optimize design of the variable. In DQ2, we took both an enrollment-focused approach and a progression-focused (the main study outcome) approach. We plotted increasing thresholds for low eGFR in the cohort against the proportion of the cohort with at least one eGFR below that threshold (Fig 4). If the threshold for low eGFR is set at 50, only 17% of the potential cohort for one institution had a low eGFR, whereas another institution had 43% of their potential cohort meet the threshold, with a median of 33%. This suggests either significant unappreciated stratification of the population across institutions, or significant differences in testing practice or result availability across institution. While full investigation of causes for heterogeneity was not feasible, the study team can account for the potential bias in cohort selection or sensitivity analyses (Theme: Comparatively anomalous institutional clinical values). We also attempted to evaluate the plausibility of

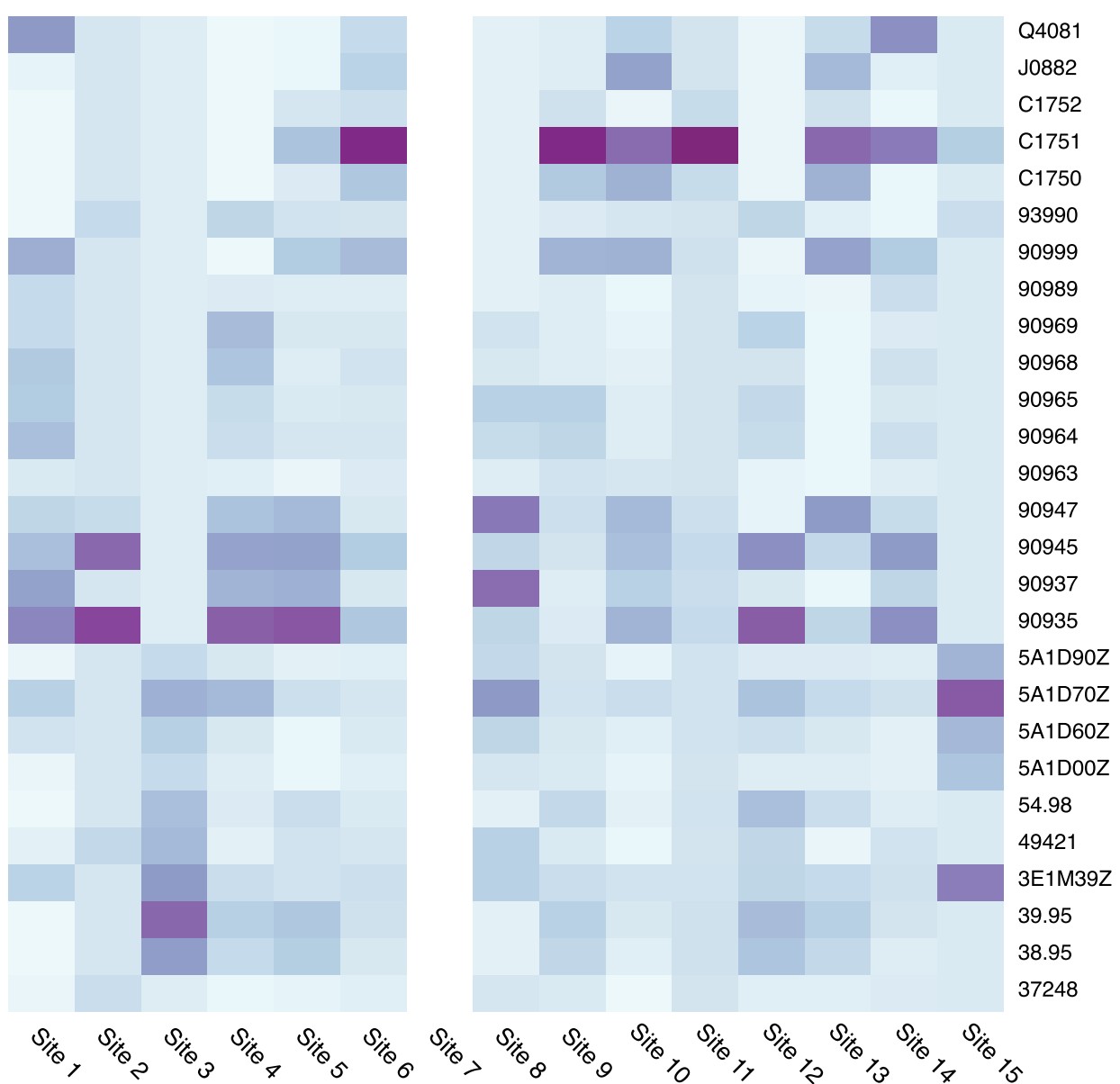

**Fig 3.** *Data Quality Theme: Variation in code utilization: Dialysis procedure code frequency by site.* The shading represents the distribution of codes, with a darker shade indicating that a larger proportion of patients in the cohort have the code. To limit skewing by high utilization, one code per calendar year for each patient was used. Procedure codes with a prevalence of less than 0.5% across sites are excluded. Site 7 has no chronic dialysis codes recorded in their EHR.

other lab result data (Theme: Anomalous distribution of results) as shown in Fig 5, focusing on urine protein measurements because of its complex representation in source systems. For example, site 9 contains peaks for quantitative values of urine protein data at the values approximating 60, 100, and 150, which are likely attributable to the conversion of a qualitative result to a quantitative value, rather than a true quantitation of urine protein. Further, the majority of site 2's urine protein values are very high, which may indicate selective testing of their population.

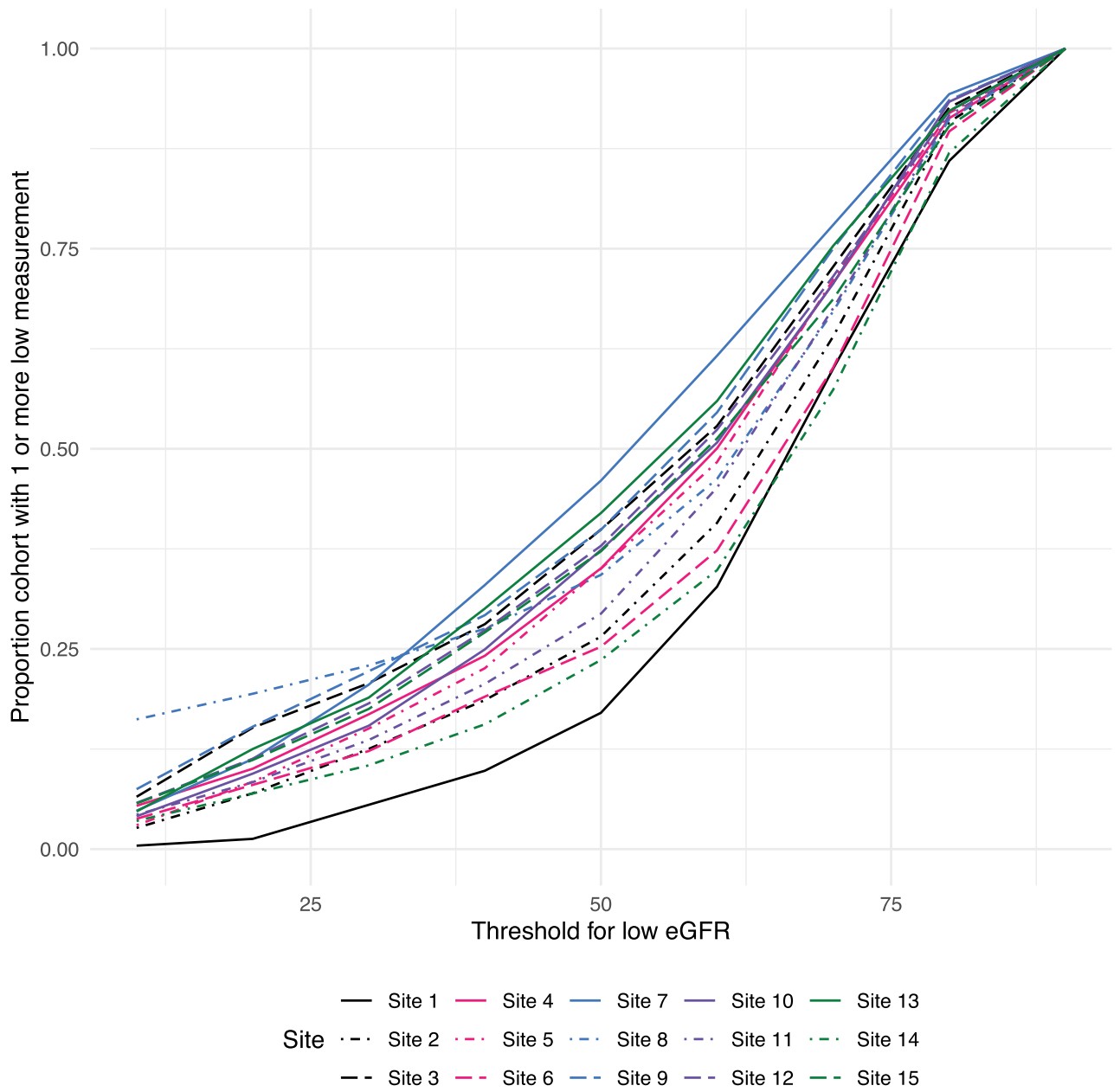

**Fig 4. Data Quality Theme: Comparatively Anomalous Institutional Clinical Values: Severity of CKD.** The graph shows the proportion of patients in the CKD cohort with at least one eGFR below the threshold shown on the x-axis.

## Resolution And Improvement

Data quality summaries were provided to each site with the twin goals of facilitating diagnosis of issues and calibrating priorities for resolution. Because PRESERVE was a retrospective study, gaps could be improved only through correction of ETL errors or discovery of additional data in source systems. Summaries were derived from the issue catalog, described in Methods. Table 4 demonstrates a selection of issues provided to one institution. This example indicates in issue 2 that this site had a relatively high number of patients with eGFR values

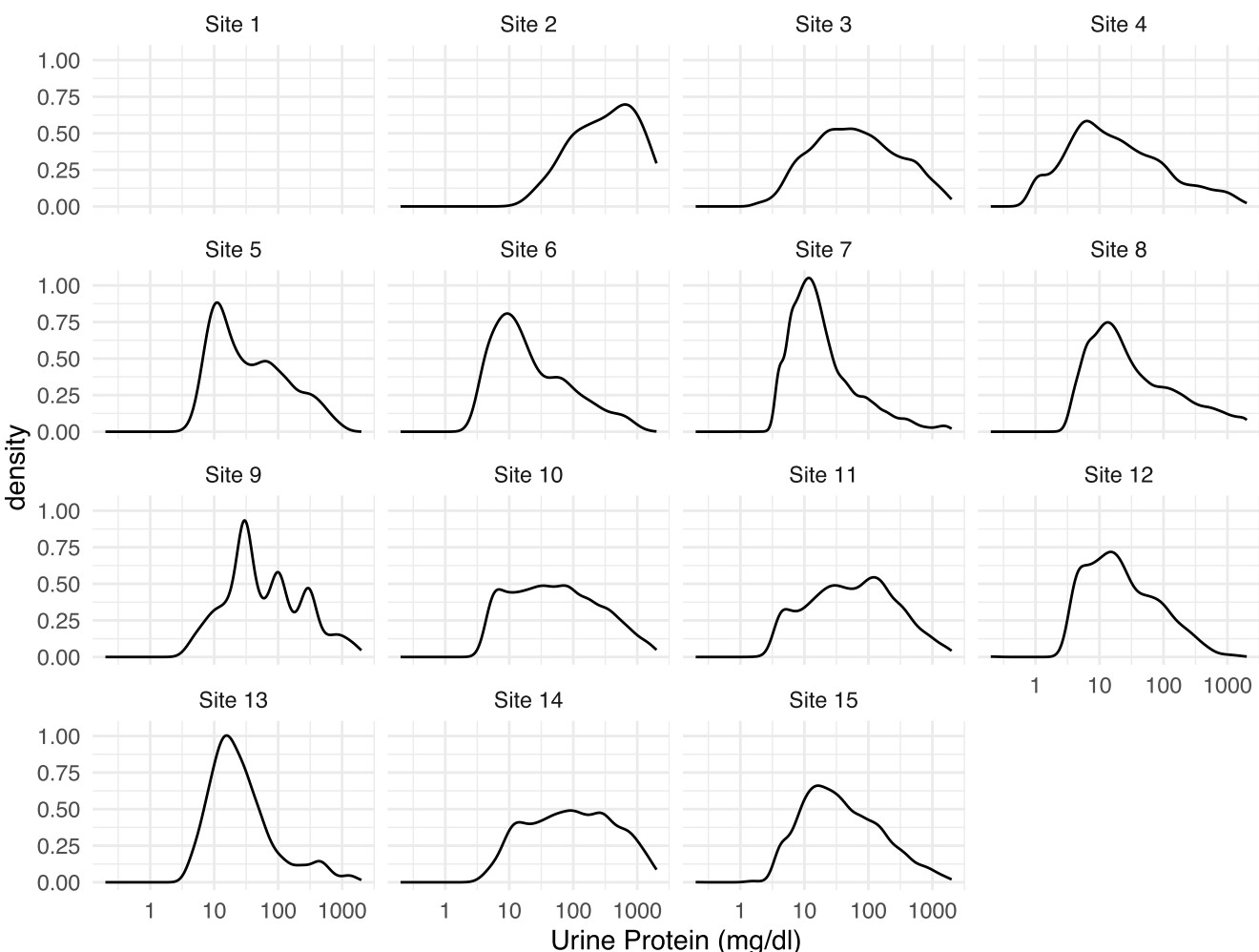

**Fig 5.** *Data Quality Theme*: *Anomalous Distribution of Results*: *Urine protein quantitative test results by site.* The graph shows the relative density of urine protein value, an indicator of kidney dysfunction in some conditions producing CKD.

within the low value range (30 - <90 mL/min/1.73m$^2$), used as an eligibility criterion. Upon further investigation, we observed that the site also contained a disproportionately large number of height measurements with low values, as indicated by issue id 141, and we concluded that unit conversions and faulty programming during data extraction could be affecting both variables, which we denoted in the *info* column in both rows.

We evaluated data quality resolution by priority status in both DQ1 and DQ2, as shown in Fig 6. In DQ1, urgent issues, which were all resolved, were related to critical cohort attrition variables such as missingness of nephrology specialty and low number of heights available to compute eGFR, other issues such as mismapping of Hispanic ethnicity so that these patients were excluded from the cohort, and missing blood pressure values. The improvement in attrition following resolution is reflected in Fig 7. Steps 4 and 8 of the attrition process require height measurements and in-person visits with nephrology providers, respectively. Affected institutions were able to identify missing data after additional discovery effort in clinical source systems, and to correct mapping issues in ETL code, in order to include them in the study data. Panel B shows significantly improved attrition following resolution of urgent priority

**Table 4. Tailored Site Data Quality Summaries.**

| unique issue id | priority | clinical or data domain | data element | dq issue | info | report reference |
|---|---|---|---|---|---|---|
| 2 | urgent | cohort definition | attrition step 5 | counts high | see also domain: anthropometrics, element:height | Fig 3.2 |
| 141 | urgent | anthropometrics | height | values low | see also domain: cohort_definition, element: attrition_step_5 | Fig 4.53 |
| 28 | high | laboratory tests | serum creatinine | missingness | calendar years missing: 2014 | Table 4.5 |
| 13 | medium | cohort definition | high serum creatinine cohort | counts low | NA | Fig 3.6 |
| 16 | medium | cohort entry | cohort entry | missingness | calendar years missing: 2009, 2010, 2011, 2012, 2013 | Fig 4.1 |
| 51 | medium | laboratory tests | quantitative urine protein measurement | missingness | calendar years missing: 2014 | Fig 4.10 |
| 137 | medium | anthropometrics | height | counts high | NA | Fig 4.51 |
| 142 | medium | anthropometrics | weight | counts high | NA | Fig 4.56 |
| 148 | medium | blood_pressure | diastolic | counts high | NA | Fig 4.61 |
| 155 | medium | blood_pressure | systolic | counts high | NA | Fig 4.64 |
| 39 | low | laboratory tests | serum cystatin | missingness | calendar years missing: 2014, 2015, 2018, 2019 | Table 4.7 |
| 79 | low | medications | calcium channel blockers | counts high | NA | Fig 4.26 |
| 83 | low | medications | loop diuretics | counts high | NA | Fig 4.29 |

issues. In DQ2, urgent priorities were related to disproportionate number of vital sign measurements or inpatient visits, missing urine protein data, missing geographic data, and disproportionately low Asian representation; all were resolved, save the last which accurately reflected the local population. The addition of geographic data was often the result of local governance allowing inclusion of geocodes in study cohort data after more extended review, while the other issues were resolved again through data discovery and improvements in mapping of terms as part of the ETL for the study cohort. Of note, because this ETL is part of the process by which regular refreshes of the PCORnet CDM data are constructed, improvements triggered by this process can persist for use in other studies.

Of the 115 identified data quality issues in DQ1, 57 (50%) were resolved by institutions by adjusting ETL from the source data. A large portion of remaining issues (22%) were reflective of EHR architecture or implementation (e.g., delay of clinical domains due to EHR unavailability), source data (9%, *e.g.*, poor capture of clinical metadata like specimen source), or clinical practice (10%, *e.g.*, greater preference for a specific antihypertensive medication). The label of *reflective of clinical practice* was assigned when the response by the site specifically mentioned a characteristic of their clinical practice as the source of the anomaly. The Coordinating Center made adjustments to resolve 10% of issues, such as expanding codesets to be inclusive of site heterogeneity. In contrast to DQ1, only 34% of issues in DQ2 were institutional improvements to data, and a larger proportion were high (46% vs 23%) or medium (47% vs 23%) priority, rather than low (2% vs 46%), reflecting the focus of DQ2 on deeper investigations of important study variables. As a result, major improvements were made to incorporate or improve variables critical to analysis, such as urine protein results (*e.g.*, increase from < 5% of the cohort to 73% with a valid result), inpatient visits (*e.g.*, deduplicated visits resulting in patterns more aligned with other sites), and chronic dialysis (*e.g.*, from missing to more complete capture). The most significant improvement was made for patient address ascertainment, reflected in Fig 8, which the study uses to ascertain area-level measures of social influences on health, including the Area Deprivation Index [31], a composite measure integrating at the census block group level multiple social influences on well-being. Also in contrast with DQ1, the plurality of data quality issues reflected source data (38% vs 9%), such as different patterns of

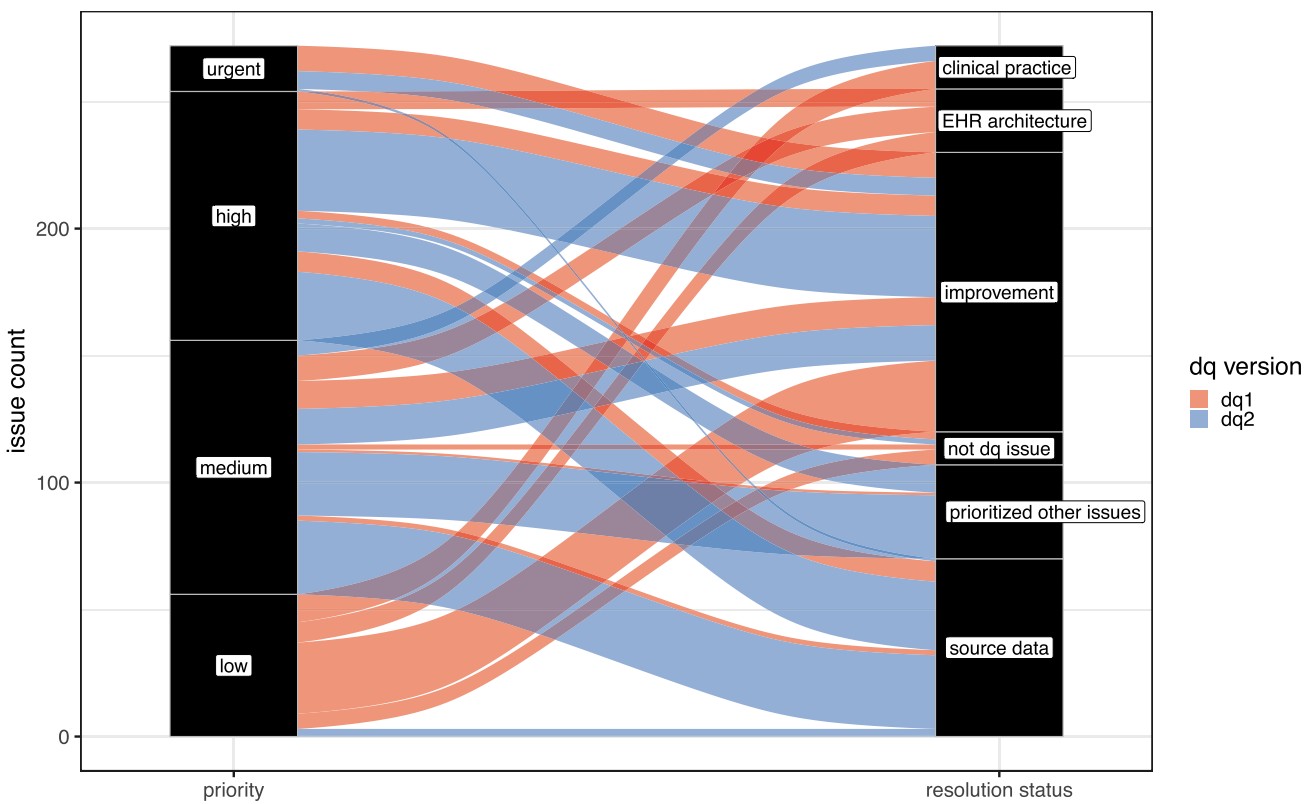

**Fig 6. Resolution of Data Quality Issues by Priority.** The Sankey diagram shows how DQ issues in each priority stratum were resolved. "Clinical practice" indicates that the issue was determined to be an accurate reflection of variation in clinical practice, while "EHR architecture" denotes limitations in data capture or ETL due to EHR design, such as restrictions in allowed values. "Source data" issues were accurate extraction of the data in clinical source systems that for other reasons did not accurately reflect patient status. By contrast, "not dq issue" indicates reported issues that were due to incorrect logic in DQ checks. "Improvement" designates issues where data quality was significantly increased across DQ cyles, while "prioritized other issues" indicates that the institutional informatics team believed the data could be improved but study timeline and resource constraints could not accommodate it.

utilization, heterogeneity in patient case-mix, and inability to access patient address history. Variation in utilization patterns included both visit type classification (*e.g.*, outpatient vs administrative visits) as well as recording of clinical events (*e.g.*, types of procedures extracted). Issues specifically attributed to clinical practice were relatively rare (4%), but were important to note. For example, in Fig 9, site 10 was the only institution where the majority of patients received an antihypertensive prior to cohort entry (theme: Event sequencing anomalies). Availability of row-level data allowed us to determine this was due to a spike in prescribing shortly prior to entry, rather than a variety of start times, suggesting little variability in clinical practice internally. Many more issues in DQ2 could not be resolved (23%), likely because of the complexity involved in investigating them, which may require a multidisciplinary team reviewing for face validity, and operational and technical teams providing input on backend systems. The three (2%) low priority issues were related to disproportionate numbers of Hispanic children at institutions. For data elements such as this, where value sets are limited and largely congruent across institutions, anomalies other than complete mismatch are more suggestive of differences in source data than of divergence in ETL processes. While the three sites here were ultimately able to correct their mapping of ethnicity values, detection of such differences is important to determine whether inconsistent classification is part of the analytic data. Further investigation may demonstrate that anomalies arise from external factors such as

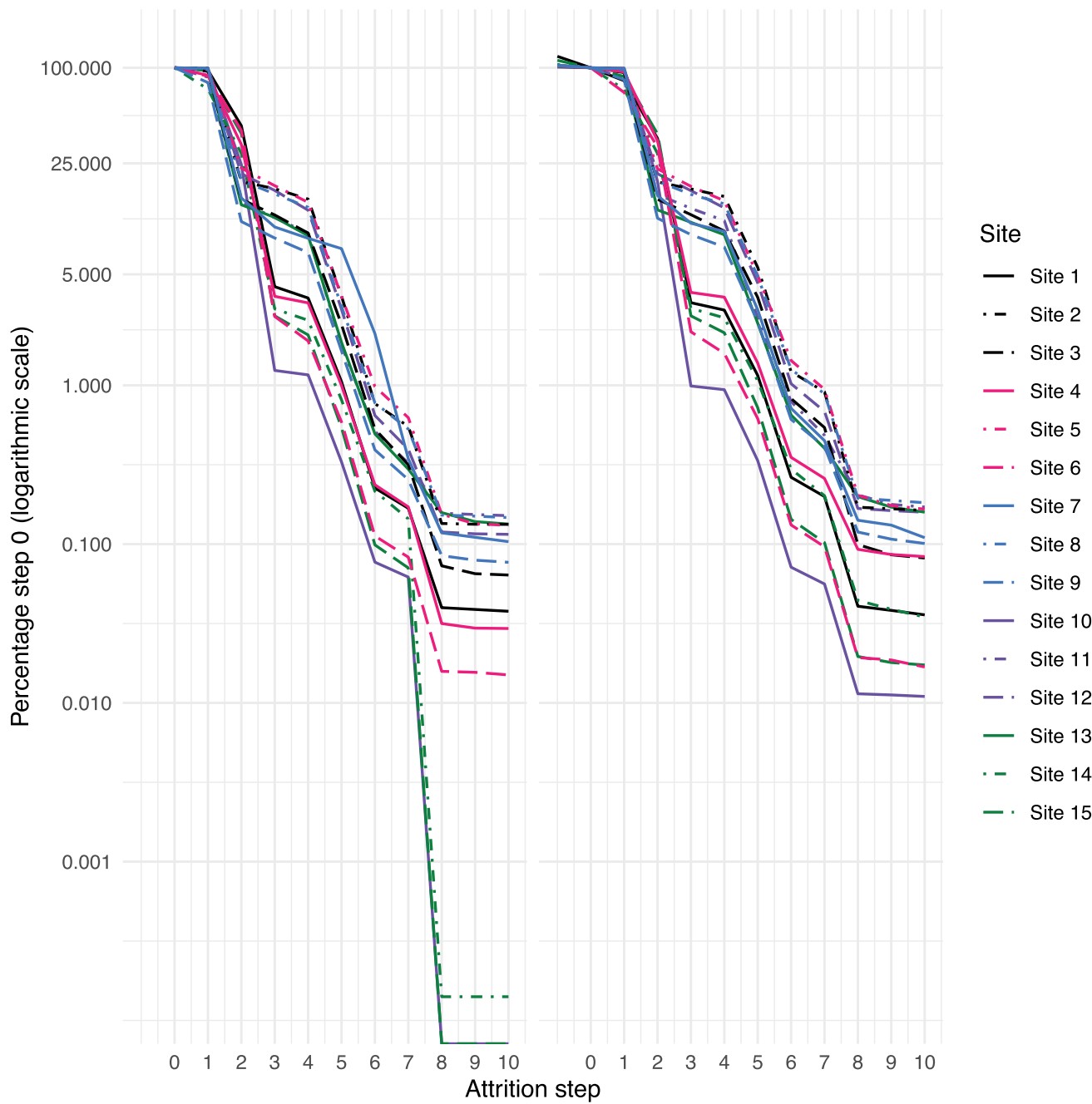

**Fig 7.** *Issue Resolution for PRESERVE Cohort Eligibility Criteria.* Percent of total patients at a site retained with each step in PRESERVE attrition criteria, shown in logarithmic scale. Panel A: DQ assessment of original data. Panel B: DQ assessment of improved data, showing impact of specialty resolution for 2 sites. Attrition steps were applied sequentially: 0 –any encounter in the study eligibility interval (2009–2021); 1 –any in-person visit in the study eligibility interval; 2 –at least one serum creatinine value; 3 –age 1–18 years at time of serum creatinine measurement; 4 –height measurement within 90 days of eligible serum creatinine measurement; 5 –at least 1 eGFR value between 30 and 90 ml/min/1.73m$^2$; 6 –two additional eGFRs in this range, at least 90 days apart; 7 – no intervening normal ($\geq$90) eGFR between two qualifying low eGFRs; 8 –any in-person visit with a nephrologist; 9 –no chronic dialysis procedure codes prior to cohort entry; 10 –no kidney transplant procedure codes prior to cohort entry.

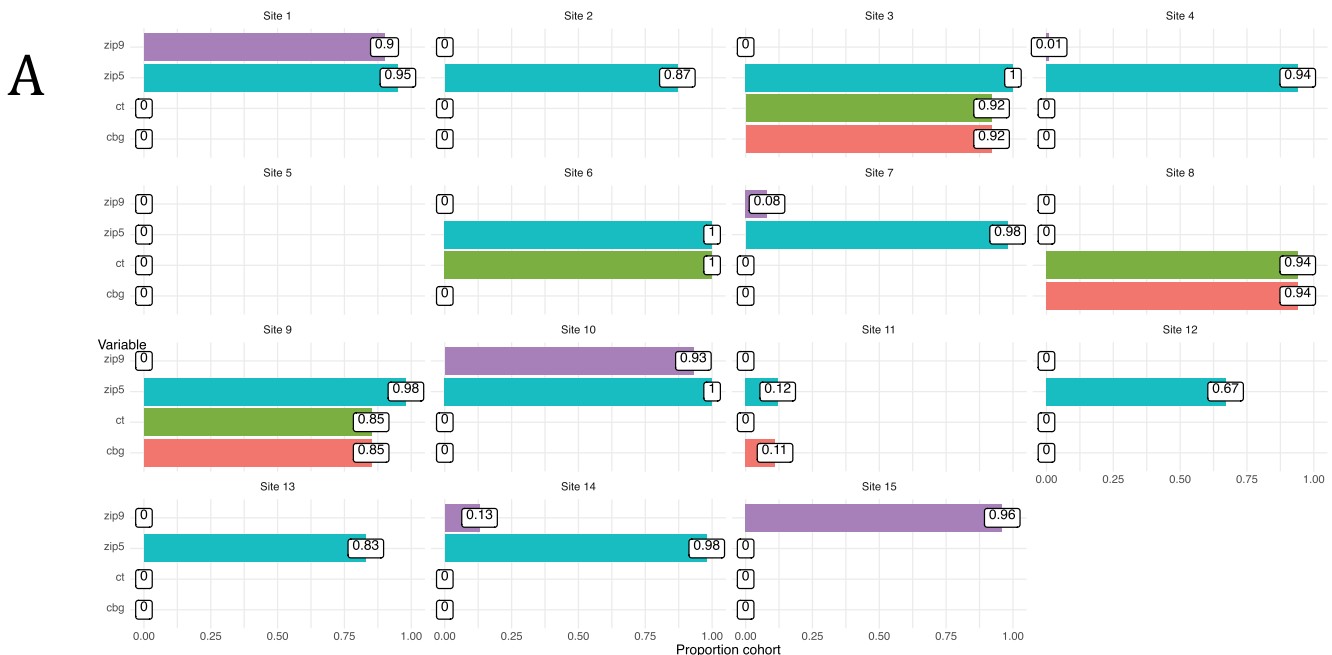

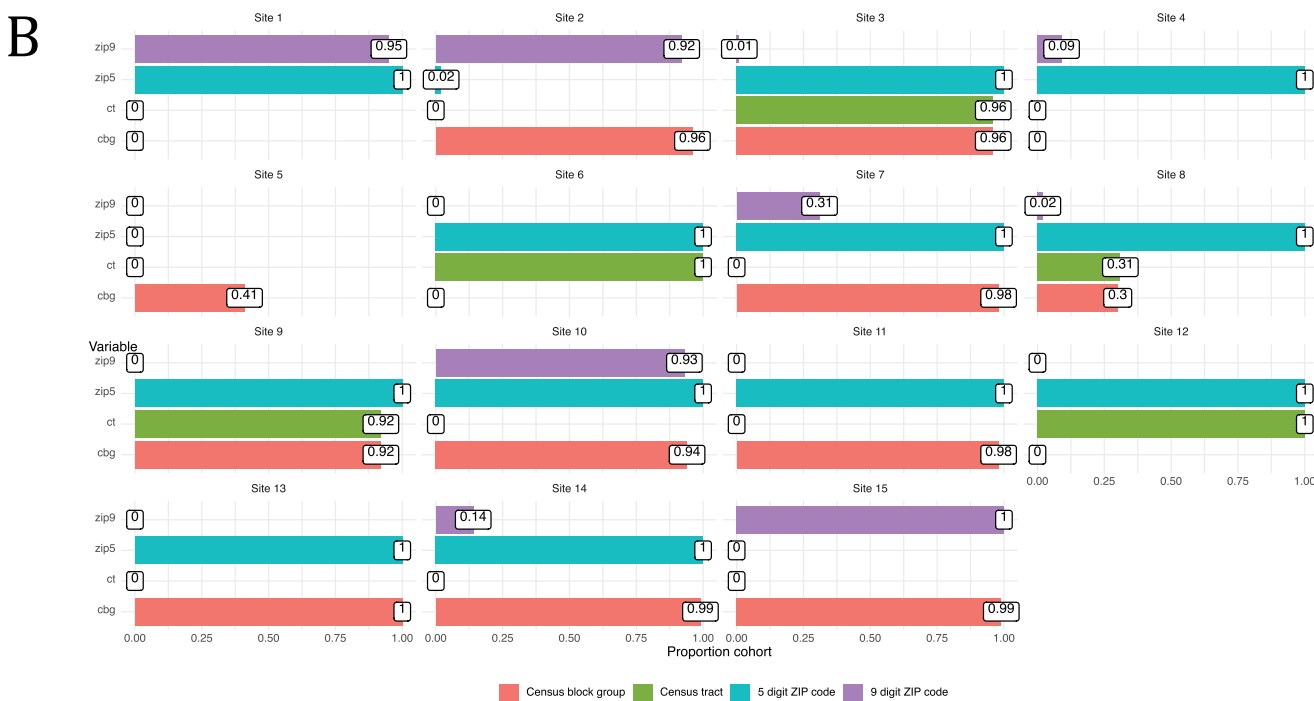

**Fig 8. Issue Resolution for Geographic Location.** Bars show the proportion of the cohort having the indicated type of location data (5-digit ZIP, 9-digit ZIP, census tract, census block group). Panel A: DQ assessment or original data. Panel B: DQ assessment following resolution, showing improvement in geographic data used in PRESERVE (shown as increase in red bars denoting census block group granularity). Note also the loss of census location for site 8, where data were available on site in DQ1, but could not ultimately be shared for study use due to institutional policies.

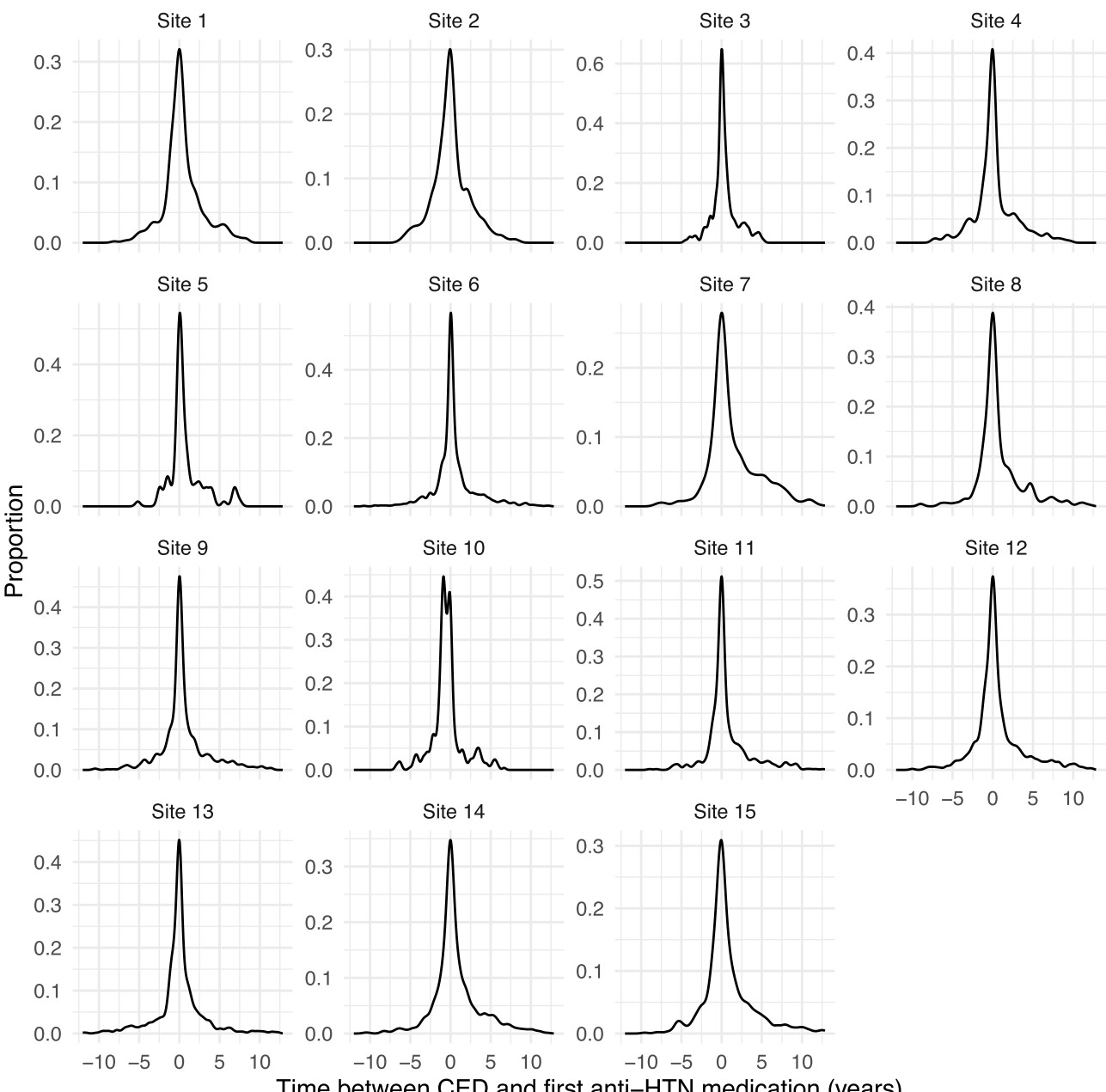

**Fig 9. Further Investigation of Antihypertensive Medication Start (Event Sequencing Anomalies).** The differences between cohort entry and initiation of antihypertensive medication are shown for the cohort at each PRESERVE site.

differences in data collection or inherent limitations of the value set [32], rather than resolvable technical problems, which may require changes to the analytic approach.

## Discussion

Systematic data quality assessment focused on fitness for a study's analytic requirements can identify threats to validity prospectively, allowing them to be addressed in data preparation or

analytic design. The purpose of this study was to refine a framework for this systematic approach [26], classifying methods for data quality evaluation familiar to informaticists and clinical researchers. It is beyond the scope of any single study to systematically develop a complete variable-agnostic catalog for all study-specific check applications. Rather, we focused on validating the framework and demonstrating its utility in real world data, using two rounds of DQA as part of the PRESERVE multisite observational study in. Of note, all data contributors had successfully completed PCORnet data curation, highlighting the importance of adequate SSDQA to address study-specific requirements. The aggregate level round (DQ1) interrogated key variables to drive development of the study protocol, focusing on missingness, implausible values, and cohort eligibility checks. Causes for identified issues ranged from ETL errors, such as duplication of rows, to unexpected characteristics of the underlying clinical data, such as defaulting lab values to implausibly high numbers. In some cases institutional data appeared anomalous due to a lag in deploying EHR systems. We also examined heterogeneity in clinical coding for the key procedures of chronic dialysis or renal transplant, to guide clinical investigators in adjusting study analytic plans appropriately. Finally, DQ1 led to improved cohort construction across institutions, particularly through improvement in ETL of necessary provider specialty data.

The goal of the row-level round (DQ2) was to implement a more detailed set of data quality checks on the data that would be used for study analyses. We incorporated checks that estimated uniformity of the primary outcome measure (eGFR) across institutions, temporal relationships between cohort entry and key CKD-related facts such as number of urine protein measurements or initial antihypertensive medications, correlation between systolic and diastolic blood pressure (the study's primary independent variables), and utilization of specialist care to ascertain comorbidity burden in the cohort. Because of the greater granularity than DQ1, the checks often highlighted factors that impact statistical analyses but were not errors in ETL. For example, institutional heterogeneity in patient case-mix was an identified issue that investigators must consider during analyses. Importantly, EHR data may not always align with expectations of study investigators based on idealized care. For example, the stability of patients with normal eGFR and the relatively high proportion of patients across all institutions who were on an antihypertensive medication prior to cohort entry both suggested that eligibility criteria would require additional refinement.

The SSDQA reports were designed to be usable and informative, while complementing the standard PCORnet data curation process. Rather than simply listing metrics, we produced visualizations and detailed descriptions of quality findings with notation of outlier institutions. This effort made results more interpretable to both institutional ETL analysts and investigators. We managed issue investigation via site-specific GitHub repositories, where institutions could provide feedback and updates on efforts at resolution. These efforts produced significant improvements; though it required investment of resources, the work also produced clear documentation of data quality. Most significantly, we developed a taxonomy and classification of study-specific checks that are reusable tools and can be replicated across different contexts and domains.

We note several important limitations of the current work. First, SSDQA requires significant effort, which must be allocated in the study's planning. Frameworks such as the one applied here streamline check selection and process optimization, but checks must be built and interpreted, as must the interactions between them. The effort yields value by detecting issues earlier and driving improvements, rather than analyses being stalled or results being compromised by unexpected problems. Nonetheless, further work is needed on methods to reduce the opportunity cost, whether through additional reuse, automation, or other mechanisms. Ultimately, developing reusable resources is an ongoing process, necessarily balanced

against need to tailor programming effort to the current use case. Second, the study's timeline for data specification, extraction, DQ testing, and issue resolution did not permit institutional teams to investigate all issues or allow for extended collaboration across different teams to address the results. This limitation was balanced by defining priorities in collaboration with the study analysis team, so that institutions could focus their effort on the most significant data quality issues. Third, it was challenging to comprehensively pre-specify SSDQA checks prior to completion of all analyses plans and variable definitions. Future work may benefit from interleaving later portions of SSDQA, particularly those likely to produce results that alter data analysis rather than ETL, throughout the course of a study rather than limiting to discrete waves. In this way, early analytic decisions can prompt the application of checks aligned with evolving analysis plans, to rigorously interrogate the quality of data. Fourth, SSDQA is inherently multidisciplinary, involving collaboration across multiple areas of expertise and institutional roles to interpret findings. This study focused institutional dissemination of results to ETL analysts, who often needed additional *ad hoc* consultation with local domain experts to address issues. Future studies can incorporate into timelines the need to facilitate discussion continuously between domain experts, study investigators and analysts, and ETL programmers to respond to changing study design and analytic requirements resulting from the SSDQA output.

In comparison to typical assessment limited to missingness and overall facial plausibility, this work demonstrates the added value of implementing *fit for use* data quality assessment using a systematic approach. It complements current practice in the statistical literature to control for data anomalies through analytic methods such as adjusting for confounders or reducing bias from misclassification [33,34]. The purpose of this work is to deepen our understanding of the characteristics and representation of a particular dataset to make better and informed decisions about study design and subsequent analyses. For example, uncovering anomalous concept-set distributions may prompt investigators and statisticians to evaluate the tradeoffs of altering variable definitions, creating new concept sets, or imputing for missing values. As a result of this approach, we resolved several urgent data quality issues that would have otherwise been highly disruptive during the analytic phase of the study. In addition to the improved data quality for this particular study, the resolved issues ultimately improved institutions' research data as a whole. The framework we applied is not restricted to a particular data model or type of study. The principles and process can be applied to many research study designs, and check types can be cataloged and reused for future studies and shared with other researchers. Future efforts must focus on automating processes via tools such as software packages that apply pre-specified combinations of check types and data quality probes across a variety of contexts. It will also be important to refine metadata and DQ terminologies, aligning with FAIR principles to make data more findable and accessible, and to develop a robust user-interface for study teams to make decisions about check application and evaluation for their studies.

## Supporting information

**S1 Table. Data quality check catalog for DQ1 analysis.**
(DOCX)

**S2 Table. Data quality check catalog for DQ2 analysis.**
(DOCX)

**S3 Table. Considerations for issue remediation.**
(DOCX)

## Acknowledgments

The authors wish to express their gratitude to the patients and families at participating institutions whose clinical data form the basis for the PRESERVE study, as well as the efforts of treating clinicians, study investigators, and informatics team members at each institution and at the Coordinating Center who have collaborated on data acquisition and testing. The content of this manuscript is that of the authors, and does not reflect the positions of PCORI or the participating institutions.

## Author Contributions

**Conceptualization:** Hanieh Razzaghi, Amy Goodwin Davies, Christopher B. Forrest, L. Charles Bailey.

**Data curation:** Hanieh Razzaghi, Amy Goodwin Davies, Samuel Boss, H. Timothy Bunnell, Elizabeth A. Chrischilles, Kimberley Dickinson, David Hanauer, Yungui Huang, K. T. Sandra Ilunga, Chryso Katsoufis, Harold Lehmann, Dominick J. Lemas, Kevin Matthews, Eneida A. Mendonca, Keith Morse, Daksha Ranade, Marc Rosenman, Bradley Taylor, Kellie Walters, L. Charles Bailey.

**Formal analysis:** Hanieh Razzaghi, Amy Goodwin Davies, Samuel Boss, Kimberley Dickinson, K. T. Sandra Ilunga, L. Charles Bailey.

**Funding acquisition:** Michelle R. Denburg, Christopher B. Forrest.

**Investigation:** Hanieh Razzaghi, Amy Goodwin Davies, Samuel Boss.

**Methodology:** Hanieh Razzaghi, Amy Goodwin Davies, Yong Chen.

**Project administration:** L. Charles Bailey.

**Supervision:** Hanieh Razzaghi, H. Timothy Bunnell, L. Charles Bailey.

**Visualization:** Hanieh Razzaghi, Amy Goodwin Davies, Kimberley Dickinson.

**Writing – original draft:** Hanieh Razzaghi, L. Charles Bailey.

**Writing – review & editing:** Hanieh Razzaghi, H. Timothy Bunnell, Yong Chen, Elizabeth A. Chrischilles, Kimberley Dickinson, David Hanauer, Yungui Huang, K. T. Sandra Ilunga, Chryso Katsoufis, Harold Lehmann, Dominick J. Lemas, Kevin Matthews, Eneida A. Mendonca, Keith Morse, Daksha Ranade, Marc Rosenman, Bradley Taylor, Kellie Walters, Michelle R. Denburg, Christopher B. Forrest, L. Charles Bailey.

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
