## [Decision Letter · Decision Letter 0]

19 Dec 2023

PDIG-D-23-00422

Systematic data quality assessment of electronic health record data to evaluate study-specific fitness: report from the PRESERVE research study

PLOS Digital Health

Dear Dr. Razzaghi,

Thank you for submitting your manuscript to PLOS Digital Health. After careful consideration, we feel that it has merit but does not fully meet PLOS Digital Health's publication criteria as it currently stands. Therefore, we invite you to submit a revised version of the manuscript that addresses the points raised during the review process.

Please submit your revised manuscript within 60 days Feb 17 2024 11:59PM. If you will need more time than this to complete your revisions, please reply to this message or contact the journal office at digitalhealth@plos.org. Please include the following items when submitting your revised manuscript:

We look forward to receiving your revised manuscript.

Kind regards,

Miguel Ángel Armengol de la Hoz, Ph.D.

Section Editor

PLOS Digital Health

Journal Requirements:

1. We ask that a manuscript source file is provided at Revision. Please upload your manuscript file as a .doc, .docx, .rtf or .tex.

Additional Editor Comments (if provided):

Reviewers' comments:

Reviewer's Responses to Questions

**Comments to the Author**

1. Does this manuscript meet PLOS Digital Health’s publication criteria? Is the manuscript technically sound, and do the data support the conclusions? The manuscript must describe methodologically and ethically rigorous research with conclusions that are appropriately drawn based on the data presented.

Reviewer #1: Partly

Reviewer #2: Yes

Reviewer #3: Yes

Reviewer #4: Partly

2. Has the statistical analysis been performed appropriately and rigorously?

Reviewer #1: N/A

Reviewer #2: N/A

Reviewer #3: N/A

Reviewer #4: N/A

3. Have the authors made all data underlying the findings in their manuscript fully available (please refer to the Data Availability Statement at the start of the manuscript PDF file)?

Reviewer #1: Yes

Reviewer #2: Yes

Reviewer #3: No

Reviewer #4: No

4. Is the manuscript presented in an intelligible fashion and written in standard English?

Reviewer #1: No

Reviewer #2: Yes

Reviewer #3: Yes

Reviewer #4: Yes

5. Review Comments to the Author

Reviewer #1: This manuscript offers an evaluation of fitness of data for intended research use that was performed via a 2-round data quality assessment (DQA) of a 16-site EHR-based study of chronic kidney disease. Unfortunately, the writing and organization of the content in this manuscript was very difficult to follow. There are details that appear inconsistent, calling into question the rigor of the work and robusticity of findings reported. The methods were inadequately described (e.g., while several places refer to 2 rounds of DQA, on line 445 a third round is indicated). This might be attributable to overreliance on Razzaghi et al. 2022 when details must be addressed in the current manuscript. 

While this 16-site study might be an illustrative example for how a DQA has been performed and enable an enlightening discussion of the benefits that arise from such processes being performed, the manuscript itself (with its inconsistencies in reporting, ambiguity of methodology, and organizational problems) does not clarify how and to what extent data quality issues were resolved in this instance or how they could be in other applications of a DQA. For example, the abstract contains aspects that are not developed in the manuscript (e.g., how two institutions’ exclusion was avoided; how exactly data problems were avoided once identified; in what regard multidisciplinary teams are advantageous; etc.). 

I've included more specific feedback in the hopes you find it constructive when revising the manuscript:

1. In both Round 1 DQ and Round 2 DQ, how were the designations (i.e., urgent, high, medium, and low priority) made and by whom? How were themes identified? Was there a qualitative code book used by researchers? If more than one researcher made the designation, how were discrepancies handled? If the “related to” sentences appearing on p18 are about the data quality designation, it’s unclear why that is being discussed in a section labeled “Data Quality Remediation” (which implies corrective action following identification of a data quality issue). In sum, the methods are not adequately explained or, at best, are oddly relegated to the end of the paper. For example, it is only at line 445 that readers suddenly see there was a Round 3 performed. The two supplemental tables of “catalogs of checks and accompanying annotation” are not clearly explained as to how they were created, applied, or interpreted. 

2. Fig. 1 displays only 12 sites, Fig. 2, Fig. 3, Fig. 6, and Fig. 7 display only 15 sites, and Fig. 4 displays 14 sites; however, the description of the PRESERVE study is that there are 16 sites. Only Fig. 5 displays 16 sites. Why is there a discrepancy? This should be clarified somewhere (whether in a figure caption or discussion appearing in the main text).

3. Table 2 has confusing column headers and content - particularly when one recognizes that readers will not yet have seen the methods buried at the bottom of the manuscript. 

4. What do the authors mean by “misclassified ethnicity” at line 243 p19? Misclassified by whom, in what way, and when? How are population identifiers and their inconsistencies addressed? While the 2023 NASEM report on population identifiers was focused on genetic and genomic research uses, the definitional clarifications are relevant and important considerations with regard to EHR data’s fit-for-purpose as well, particularly when one acknowledges that ethnic identity (if you are talking about patient self-reported ethnicity) develops and changes over individuals’ lifetimes and also, distinctly, the social constructs change over time (see, e.g., the PEW Research chart of Feb. 6, 2020 clarifying how the US Census has altered its handling of race and ethnicity over time at https://www.pewresearch.org/interactives/what-census-calls-us/). For example, did you examine how, precisely, Meaningful Use Stage 2 (which involved the 1997 OMB minimum categories) was implemented/achieved at each study site, when that occurred at each site, and how that affects the data quality? 

5. Aspects of the discussion really should appear much earlier in the manuscript for readers to understand what is being done and why (e.g., the text at lines 291-296). 

6. There is inadequate information about how data quality issues identified were corrected/resolved by the various institutions. Who assessed whether and how were changes determined to be sufficient to make the data appropriate for subsequent analysis? 

7. Fig 8 is not mentioned anywhere in the main text. Did you intend for the reference to “Figure 1” on line 383 to direct readers to Fig 8 instead?

There are an assortment of writing issues that are confusing and/or distracting and should be addressed with any revisions. These include the following:

8. Line 87, p6 refers only to use of data collected for “operational” purposes, but EHR data actually also include data collected for “treatment” and “payment” purposes (see HIPAA 164.502(a)(1)(ii) “for treatment, payment, and healthcare operations”). I would suggest you rephrase to be more precise and accurate.

9. Line 99, p6 requires a period after “et al” so it accurately communicates the abbreviation of the latin phrase meaning “and others.” This problem is repeated throughout the manuscript (e.g., Line 129, p8).

10. It’s unclear why the words “completeness,” “plausibility,” or “conformance” are italicized in Lines 99-100 p6 and seems unnecessary. Same with “fit-for-use” at Line 101. The italics font does not appear to provide any extra clarity. Same is true elsewhere in the manuscript (e.g., words on lines 130-133, p8). The only place where italicized font seems to provide clarity is when you use it for mentions of “Round 1 DQ” (such as on line 140, p8) and “Round 2 DQ” (such as on line 149, p9). However, you do not use this style consistently (e.g., “Round 2” is not italicized on line 156, p9; inside Table 1; or at lines 272-273, p21). This manuscript overutilizes italics, which is likely to create confusion for readers. For example, why is serum cystatin italicized in some places but not italicized in Fig. 1?

11. For lines 141-145, p8, it is unclear what the numbers in the parentheses of the 10 listed domains mean. Are those references to citations or perhaps some sort of count that relates to the “79 data quality checks” noted in line 141? If citations, there are missing references from the list at the end of the manuscript (which only goes to Ref #24). It appears that they might be counts that sum to 79; however, if that’s the case, the sentence structure is highly problematic/confusing when it continues to include “four check categories” that also have parenthetical numbers listed. Even though the next sentence begins with “as the counts indicate,” the phrasing of the preceding sentence is clear as mud. The same criticism applies to how you’ve structured lines 150-156 on p9.

12. USRDS is never spelled out as the United States Renal Data System. That should be done in the first appearance of the acronym, which seems to be Line 168, p9. 

13. “ETL” (e.g., line 281, p22) is a term of art that should be explained for readers. 

14. The 73-word sentence at lines 190-195 is meandering and confusing. Consider breaking it up for readability. 

15. Sometimes you call the first round of data quality assessment “Round 1 DQ” and later “aggregate level round (DQ1)”. Is there a rationale for when you use which descriptor? A similar question applies to reference of “Round 2 DQ”; “row-level phase (Round 2 DQ)” at line 234, p18, and later “record-level round (DQ2)” at line 313, p23.

16. What is the purpose for use of the opaque term “stakeholders” at line 350? Is there a more appropriate and less value-laden term available? For examples, see alternative terms discussed by the CDC at https://www.cdc.gov/healthcommunication/Preferred_Terms.html

Other issues that copyeditors should help correct include these:

17. It is unclear why the beginning of each paragraph is not indented properly. 

18. Extraneous space following the hyphen in “healthcare-specific” in the abstract at line 46, p3

19. Unnecessary commas following “governance” in line 85, p6; following “overall” in line 110, p7; following “elements” in line 185, p13; following “cohort” in line 189, p14

Reviewer #2: Your article excellently demonstrates the significance of robust data quality testing, especially in studies making secondary use of clinical data. The systematic approach you implemented for data quality assessment, integrating widely adopted concepts with healthcare-specific evaluation methods, was particularly noteworthy. The two rounds of evaluation you conducted, focusing on both high-level aggregate evaluation and extended row-level data analysis, provided a comprehensive understanding of the data quality issues encountered.

The identification of over 100 data quality issues in each round, encompassing various dimensions such as completeness, data model conformance, cross-variable concordance, consistency, and plausibility, is a substantial contribution. Your emphasis on cataloging and prioritizing issues for remediation, leading to the successful resolution of numerous data gaps and extraction errors, is commendable. Moreover, your insights into complexities related to measures of kidney function and their implications for outcome definitions provide valuable insights for future research in this domain.

Your study rigorously evaluated the fitness of data for its intended use, and the framework you have developed is both reusable and theoretically robust. The findings from your work not only contribute significantly to addressing data quality issues in healthcare research but also underscore the imperative need for multidisciplinary teams when handling real-world data.

Reviewer #3: Summary of the Research and Overall Impression:

Razzaghi et al.'s paper, "Systematic Data Quality Assessment of Electronic Health Record Data to Evaluate Study-Specific Fitness: Report from the PRESERVE Research Study," provides a comprehensive exploration of a study-specific data quality framework in the context of a multicenter retrospective study. 

This work builds upon the author's previously cited manuscript, "Developing a Systematic Approach to Assessing Data Quality in Secondary Use of Clinical Data Based on Intended Use." 

The paper effectively aligns its abstract and tables with the primary study objective.

Strengths:

This manuscript offers significant strengths by highlighting the critical role of Data Quality Assessment (DQA) in real-world clinical research, particularly within Electronic Health Record (EHR)-based studies. It vividly illustrates the substantial heterogeneity among sites and inconsistencies across data variables, underscoring the importance of DQA. The incorporation of multiple figures (1-7) and table 2 serves as a compelling showcase of the framework's real-world applicability.

Overarching Comment:

While the paper successfully applies the framework, there is an opportunity to transition it from being primarily an application piece to a demonstration of the framework's potential generalization across diverse research contexts.

Minor Weaknesses:

- The paper effectively details the application and operationalization of the framework in the methodology section. However, it could benefit from clearer guidance on how readers can replicate

---

## [Decision Letter · Decision Letter 1]

15 Mar 2024

PDIG-D-23-00422R1

Systematic data quality assessment of electronic health record data to evaluate study-specific fitness: report from the PRESERVE research study

PLOS Digital Health

Dear Dr. Razzaghi,

Thank you for submitting your manuscript to PLOS Digital Health. After careful consideration, we feel that it has merit but does not fully meet PLOS Digital Health's publication criteria as it currently stands. Therefore, we invite you to submit a revised version of the manuscript that addresses the points raised during the review process.

Please submit your revised manuscript within 60 days May 14 2024 11:59PM. If you will need more time than this to complete your revisions, please reply to this message or contact the journal office at digitalhealth@plos.org. Please include the following items when submitting your revised manuscript:

We look forward to receiving your revised manuscript.

Kind regards,

Miguel Ángel Armengol de la Hoz, Ph.D.

Section Editor

PLOS Digital Health

Journal Requirements:

Additional Editor Comments (if provided):

Reviewers' comments:

Reviewer's Responses to Questions

**Comments to the Author**

1. If the authors have adequately addressed your comments raised in a previous round of review and you feel that this manuscript is now acceptable for publication, you may indicate that here to bypass the “Comments to the Author” section, enter your conflict of interest statement in the “Confidential to Editor” section, and submit your "Accept" recommendation.

Reviewer #5: All comments have been addressed

Reviewer #6: (No Response)

Reviewer #7: (No Response)

Reviewer #8: All comments have been addressed

2. Does this manuscript meet PLOS Digital Health’s publication criteria? Is the manuscript technically sound, and do the data support the conclusions? The manuscript must describe methodologically and ethically rigorous research with conclusions that are appropriately drawn based on the data presented.

Reviewer #5: Yes

Reviewer #6: Partly

Reviewer #7: Yes

Reviewer #8: Yes

3. Has the statistical analysis been performed appropriately and rigorously?

Reviewer #5: Yes

Reviewer #6: Yes

Reviewer #7: N/A

Reviewer #8: Yes

4. Have the authors made all data underlying the findings in their manuscript fully available (please refer to the Data Availability Statement at the start of the manuscript PDF file)?

Reviewer #5: Yes

Reviewer #6: No

Reviewer #7: Yes

Reviewer #8: Yes

5. Is the manuscript presented in an intelligible fashion and written in standard English?

Reviewer #5: Yes

Reviewer #6: Yes

Reviewer #7: Yes

Reviewer #8: Yes

6. Review Comments to the Author

Reviewer #5: The author amended the manuscript based on comments from the previous reviewers. 

There are no comments from my site. 

the manuscript is well prepared and provides readers with clinically valuable information.

Reviewer #6: The manuscript effectively demonstrates the application of systematic data quality assessment in a real-world research setting. The thorough evaluation of fitness for study-specific requirements is commendable and highlights the importance of addressing data quality issues in secondary analyses of clinical data.

Besides, the manuscript provides a comprehensive analysis of the application of a study-specific data quality assessment (SSDQA) framework in the context of the PRESERVE study. However, it has several limitations and areas for improvement. These include limited generalizability, resource intensiveness, incomplete evaluation, lack of longitudinal analysis, and absence of comparative analysis.

The manuscript lacks a thorough discussion of existing research on SSDQA frameworks in the context of observational studies using electronic health record (EHR) data. A more comprehensive review of the literature could help contextualize the study's contributions and identify gaps in current knowledge or methodologies. Comparing the SSDQA framework with existing approaches in the literature would help evaluate its novelty and effectiveness.

Validation and benchmarking of the framework against established standards or best practices in data quality assessment would enhance the credibility and reliability of the findings. Future directions for the SSDQA framework could be discussed, including potential extensions or refinements, opportunities for further empirical testing, and cross-disciplinary insights from related disciplines such as epidemiology, biostatistics, and health services research.

Incorporating insights from related disciplines like epidemiology, biostatistics, and health services research could enrich the discussion and broaden the study's relevance to a wider audience. Addressing these limitations could enhance the manuscript's contribution to the field by providing a more nuanced understanding of the SSDQA framework's effectiveness and applicability in observational health data analysis.

Reviewer #7: The manuscript is of a outstanding importance for the field and it highliths the need for human activity in data quality assessment rather then only relying on the so deified anrtificial intelligence and machine learning. The authors have made a well panned and rigourous work which embasis other researcher to follow their paht and previous efforts to improve data quality assessement for the secondary analysis (and primary, in some cases too) studies that are so necessary and important both beacuse the data are available and low cost and because provides rech information and allows great researchs to be performed with.

Nevertheless, there are some minor issues for the authors to address, presented by section and lines:

Abstract:

Lines 52 and 57: The term “remediation” induces the readers to infer that the procedure to improve data quality does not correct the issues, but provides some kind of “non-conventional strategy” to make data usable. Consider using “resolution” or another synonym.

Line 64: Consider exchanging “backed by” for relies on/ conducted following.

Line 64: The sentence starting with “Major data” is confusing. Are the authors trying to say that “major data quality issues addressed in the conduction of the process would have otherwise…”?

Introduction:

Line 114: The term “programs de novo” needs to be explained once it is not usual or commonly used for this purpose, thus possibly misleading the interpretation.

Line 115: A brief explanation on what Study-specific Data Quality Assessment is crucial for the manuscript comprehension. 

Line 116: This paragraph states the aims and part of the methods. Consider presenting only the background and the directions for the study background construction.

Line 130: This paragraph would be better placed previously to the one above, which might be excluded or rewritten in a less detailed form since it presents information that will be again presented by the authors in the following section.

The authors do not present and overview of what PCROnet is, so the comprehension of the methodology and study conduction get hard for the readers that are not familiarized to it.

Material and Methods:

This section lacks an introduction, and this issue could be addressed by replacing the paragraph starting in line 116 to this section.

Line 157 and 158: There is no Figure 1 in the manuscript. 

Line 215: The sentence starting with “to address” if confusing. Did the authors develop a software? Did they followed a software development guideline/procedure to perform the study?

Line 238: What is PRESERVE CC? It is only presented in the next paragraph, so this explanation needs to be moved to this sentence.

Line 243: What is a SAS query? Is it related to SAS software?

Results:

Lines 313 to 315: the sum of the percentages results in 101%.

Line 413: What is Area Deprivation Index? Although the authors mention the reference for it, a brief explanation/presentation is required.

Reviewer #8: The manuscript presents valuable insights into systematic data quality assessment in the context of electronic health record data analysis. However, several key areas require clarification and elaboration to enhance the reproducibility and credibility of your findings. Specifically, 

Abstract: 

1- The abstract mentioned on line 53 “over 100 quality issues”, this statement lacks specificity and the method used by the authors is not specifically mentioned.

2- It was suggested on line 65 the need for muti-disciplinary teams to address data quality issue, how? Why? This statement appears a bit general and could be to some extent be ambiguous.

Introduction

1. Line 84, the citation needs to precede the end of the statement. 

2. Line 115, define the acronym DQA.

Results

1. Line 140, define DQ.

2. Section starting from Line 207: While the text discusses data quality remediation efforts, it provides limited insight into the specific actions taken to address identified issues. It mentions that some issues were remediated by affected institutions, but it does not elaborate on the nature of these remediation efforts or their effectiveness.

Discussion

1. How is the present SSDQA framework validated and applicable to any other research study? (What is the empirical evidence provided by the authors)

2. What are the costs and benefits associated with the implementation of this framework which happens to be resource-intensive?

Methods

1. How many institutions participate to PRESERVE study? (Lines 367-431)

2. What are the specific criteria used to assess data quality? What are the thresholds for determining good-quality data? What is the prioritization process (Lines 432-69)

3. How is the privacy of patient information protected? 

4. The method does not mention the validation procedure to assess the accuracy and reliability of the SSDQA. We suggest the author integrate this section into their manuscript.

7. PLOS authors have the option to publish the peer review history of their article (what does this mean?). If published, this will include your full peer review and any attached files.

**Do you want your identity to be public for this peer review?** For information about this choice, including consent withdrawal, please see our Privacy Policy. 

Reviewer #5: Yes: Vitalii Poberezhets, MD, PhD

Reviewer #6: No

Reviewer #7: Yes: 

Reviewer #8: No

---

## [Decision Letter · Decision Letter 2]

7 May 2024

Systematic data quality assessment of electronic health record data to evaluate study-specific fitness: report from the PRESERVE research study

PDIG-D-23-00422R2

Dear Dr. Razzaghi,

We are pleased to inform you that your manuscript 'Systematic data quality assessment of electronic health record data to evaluate study-specific fitness: report from the PRESERVE research study' has been provisionally accepted for publication in PLOS Digital Health.

Best regards,

Miguel Ángel Armengol de la Hoz, Ph.D.

Section Editor

PLOS Digital Health

Reviewer Comments (if any, and for reference):

Reviewer's Responses to Questions

**Comments to the Author**

1. If the authors have adequately addressed your comments raised in a previous round of review and you feel that this manuscript is now acceptable for publication, you may indicate that here to bypass the “Comments to the Author” section, enter your conflict of interest statement in the “Confidential to Editor” section, and submit your "Accept" recommendation.

Reviewer #9: All comments have been addressed

Reviewer #10: All comments have been addressed

Reviewer #11: All comments have been addressed

Reviewer #12: (No Response)

Reviewer #13: All comments have been addressed

Reviewer #14: (No Response)

Reviewer #15: All comments have been addressed

2. Does this manuscript meet PLOS Digital Health’s publication criteria? Is the manuscript technically sound, and do the data support the conclusions? The manuscript must describe methodologically and ethically rigorous research with conclusions that are appropriately drawn based on the data presented.

Reviewer #9: Yes

Reviewer #10: Yes

Reviewer #11: Yes

Reviewer #12: Yes

Reviewer #13: Yes

Reviewer #14: Yes

Reviewer #15: Yes

3. Has the statistical analysis been performed appropriately and rigorously?

Reviewer #9: Yes

Reviewer #10: N/A

Reviewer #11: Yes

Reviewer #12: Yes

Reviewer #13: Yes

Reviewer #14: Yes

Reviewer #15: Yes

4. Have the authors made all data underlying the findings in their manuscript fully available (please refer to the Data Availability Statement at the start of the manuscript PDF file)?

Reviewer #9: Yes

Reviewer #10: Yes

Reviewer #11: Yes

Reviewer #12: Yes

Reviewer #13: Yes

Reviewer #14: Yes

Reviewer #15: No

5. Is the manuscript presented in an intelligible fashion and written in standard English?

Reviewer #9: Yes

Reviewer #10: Yes

Reviewer #11: Yes

Reviewer #12: Yes

Reviewer #13: Yes

Reviewer #14: Yes

Reviewer #15: Yes

6. Review Comments to the Author

Reviewer #9: I consider the article to be of good quality and believe that it is of sufficient importance to be published in the respected "PLOS Digital Health" journal. However what seems to be the sectioning of the article that presents an inappropriate format. The method section is very large and takes up a large percentage of the paper volume. Also, I did not see the conclusion section, although it is mentioned in a consolidated form in the discussion section, but it seems that these items should be observed. I consider the lack of propriety between the parts of the manuscript to be the most important flaw of this valuable article.

Reviewer #10: The authors addressed all the comments of the reviewers and changed the manuscript accordingly.

Reviewer #11: 1. **Data Quality Assessment Setup:**

- Setting up and applying the framework for data quality assessment (SSDQA) to the PRESERVE study.

- Pairing variables with data quality terms and check types.

- Focusing on technical and methodological reproducibility and producing interpretable results.

2. **Data Quality Checks:**

- Percentage of patients with serum cystatin C value by calendar year.

- Variation in code utilization: Dialysis procedure code frequency by site.

- Comparatively anomalous institutional clinical values: Severity of CKD.

- Anomalous distribution of results: Urine protein quantitative test results by site.

3. **Data Quality Improvement:**

- Resolving issues by priority such as clinical practice and EHR architecture limitations.

- Providing enhancements in data quality across quality check cycles.

- Addressing eligibility criteria issues after assessment and improvement.

- Improving the accuracy of geographic data used in the study.

Reviewer #12: A very minor revision required: Line 274 appears to be missing a word after “PRESERVE CCC to run on the” which is followed by “and….”.

Overall, an excellent paper that clearly defines reliable and repeatable technics for study specific data quality analysis (SSDQA) that if practices at large could significantly reduce the burden on researchers by eliminating anomalous data early in the process. The example study described, has a data set that lends itself to quality analysis. How generalizable would these techniques be? With the SSDQA process as defined, was there a danger that the data quality validation techniques used that look for consistency across diagnosis, laboratory testing etc. might “smooth” out the data so that what was treated as data “noise” was actually “signal” that might have affected the study outcome? Agreement of data elements across multiple tests, such as diagnoses and medications for example or whether clinical information follows expected patterns based on prior data or subject matter expertise are essentially "filters" on the data that could remove real valid anomalies. Could this process result in a form of confirmation bias in the data used for the study? While the answer to these questions may be easy for the study in this case, there may be a need for mitigating controls for SSDQA applied to other situations.

Reviewer #13: For future works I would recommend following suggestions :-

1. It is recommended that future research include longitudinal data quality assessments to track and report changes over time.

2. Future work could explore the use of automated processes or machine/deep learning techniques to proactively identify and resolve data quality issues.

3. Conducting further comparative studies with other data quality frameworks would provide valuable insights.

Reviewer #14: It is with great pleasure that I had the opportunity to read the article entitled "Systematic data quality assessment of electronic health record data to evaluate study-specific fitness: report from the PRESERVE research study".

This is quite timely research focused on an important and interesting topic.

As a reviewer, my key consideration is the study's impact on the academic community. Specifically, I analyze how effectively the findings can be generalized beyond the immediate research context. While the study may have limitations regarding its generalizability, it compensates by exploring deeply into its subject, and for that reason alone it has value for publication. In other words, depth is adequately emphasized.

The article adheres to the IMRAD orientation/format, which is a positive aspect. However, in certain contexts, I also encounter an alternative format—Introduction, Literature Review, Methodology, Results, and Conclusion—that is more prevalent in European settings. I've noticed that authors who utilize this format tend to emphasize four key points in the conclusion section: contributions to theory, contributions to practice, limitations, and suggestions for future research. In practical terms, your Discussion section, which adheres to the IMRAD format, effectively includes the study's objectives (including DQ1 and DQ2), outlines its limitations, and offers insightful suggestions for future research. My suggestion is to dedicate significant attention to the remaining two aspects: contributions to theory and contributions to practice. Exploring deeply into how your study enriches existing literature and how it enhances the understanding of scientists and academics. Similarly, emphasize the practical implications of your research and how it can benefit practitioners in the field. Your article already highlights a positive aspect of developing a reusable taxonomy and classification system, which can be replicated across various contexts and domains. However, I recommend further exploration of both theoretical and practical contributions, as this will significantly enhance the appeal of your article to readers, including both scientists and practitioners.

Overall, I consider the authors' work commendable from a technical point of view. Therefore, my observations were to emphasize the strengths of the article, and, for this reason, my encouragement goes to a deeper exploration of the article's contributions.

Congratulations on your efforts thus far, best wishes for the revision process, and I hope everything proceeds smoothly.

Reviewer #15: The authors have addressed all comments made by reviewers in previous review rounds. My biggest concern in this type of study is the generalization limitations that the authors have already recognized and addressed in previous reviews. I have no new comments for the authors.

7. PLOS authors have the option to publish the peer review history of their article (what does this mean?). If published, this will include your full peer review and any attached files.

**Do you want your identity to be public for this peer review?** For information about this choice, including consent withdrawal, please see our Privacy Policy.

Reviewer #9: No

Reviewer #10: No

Reviewer #11: **Yes: **Noor al-deen mohammad yousef shehab

Reviewer #12: **Yes: **Ajai Sehgal

Reviewer #13: **Yes: **aasim ayaz wani

Reviewer #14: **Yes: **João Carlos Gonçalves dos Reis

Reviewer #15: **Yes: **David Restrepo
